

# Comparative Mesoscale Eddy Dynamics under Geostrophic versus Cyclogeostrophic Balance from Satellite Altimetry

Xinman Zhu[1], Yuhan Cao[1*], Linxiao Liu[2], Yigang Deng[1], Ruixiang Liu[3], Zhiwei You[4]

5     [1]School of Marine Technology and Geomatics, Jiangsu Ocean University, Lianyungang 222005, China

[2]Wuxi Ninecosmos Technology Co., Ltd., Wuxi 214062, China

[3]Lianyungang Meteorological Bureau, Lianyungang 222006, China

[4]School of Marine Sciences, Nanjing University of Information Science and Technology, Nanjing 210044, China

10     *Corresponding to: Yuhan Cao(yhcao@jou.edu.cn)

**Abstract.**Under quasi-steady conditions, excluding the influence of surface wind stress, the mesoscale meandering current should represent a balance among the pressure gradient force, the Coriolis force, and the centrifugal force. For mesoscale eddies, the nonlinear term induced owing to the local curvature of streamlines is non-negligible. Based on satellite-altimeter-derived geostrophic velocities and their cyclogeostrophic-corrected velocities, this study investigates the statistical characteristics of kinematic parameters and the differences in dynamical parameters of mesoscale eddies under two balanced frameworks, across five subregions of the North Pacific Ocean with the strongest eddy activity: the Aleutian Islands, the Kuroshio Extension, the South China Sea, the California Coastal Current, and the Hawaiian Islands. Results indicate that the cyclogeostrophic correction yields a 35.65% reduction in total eddy count compared to the geostrophic framework. Additionally, the cyclogeostrophic-adjusted currents are more likely to detect eddies characterized by larger radii and shorter lifetimes. The difference in eddy kinetic energy shows significant regional variability, with cyclonic and anticyclonic eddies exhibiting opposite biases in Eddy Kinetic Energy. Cyclogeostrophic correction tends to slow the decay rate of eddy energy. Eddies predominantly propagate westward. Under cyclogeostrophic conditions, velocity fluctuations are amplified, and the correction has a more pronounced effect on the translation speed of cyclonic eddies than on anticyclonic ones. Cyclogeostrophically derived vorticity is more stable, while the strain rate demonstrates stronger shear stability within the current field. Case studies further indicate that cyclogeostrophic correction is particularly significant for anticyclonic eddies: when curvature effects are included, anticyclonic eddies translate faster, possess stronger eddy potential energy, yet become less stable, more deformable, and more susceptible to dissipation. In contrast, cyclonic eddies move more slowly, and eddy–eddy as well as eddy–current interactions are weakened, leading to enhanced eddy stability.

## 1 Introduction

Ocean currents serve as critical pathways for the transport of mass and energy in the global ocean, 35     playing a central regulatory role in the Earth's climate system. Their variability is of fundamental



importance to global climate change (Robert and Sebille, 2021). Curved flows are ubiquitous in the ocean. Centrifugal force arises in curved ocean flows due to curvature effects. Mesoscale and submesoscale meandering motions at the ocean surface satisfy cyclogeostrophic (CGEO) balance under the combined influence of centrifugal, pressure gradient, and Coriolis forces (Shakespeare, 2016; Cao et al., 2023). Eddies are a typical form of curved flow structure ubiquitous in the ocean. They play an indispensable role in oceanic circulation, heat budget, energy redistribution, biogeochemical transport, and global climate change (Dong et al., 2014). Furthermore, the generation mechanisms, three-dimensional vertical structure, and evolutionary behavior of eddies are key to understanding changes in ocean stratification and circulation, providing crucial insights into the intrinsic dynamics of marine processes (Chelton et al., 2007; Dong et al., 2021; Yang et al., 2020). Eddies are categorized into cyclonic and anticyclonic types based on their direction of rotation. In the Northern Hemisphere, cyclonic eddies rotate counterclockwise, whereas in the Southern Hemisphere they rotate clockwise. Anticyclonic eddies exhibit the opposite behavior: clockwise in the Northern Hemisphere and counterclockwise in the Southern Hemisphere. Eddies cover a wide spectrum of horizontal scales, with radii ranging from several kilometers to hundreds of kilometers(Chelton et al., 2007;Chelton et al., 2011; Frenger et al., 2011; Tian et al., 2019). Those with a radius greater than or equal to the first baroclinic Rossby deformation radius (approximately 10–100 km) are classified as mesoscale eddies, while those with a radius smaller than 10 km but greater than the turbulent boundary layer scale (about 0.1–10 km) fall into the submesoscale category. Mesoscale eddies with coherent structures have longer lifespans and can persist in the ocean for several months to years, whereas smaller eddies are short-lived, typically lasting only from hours to a few days (Puillat et al., 2002; Ioannou et al., 2017; Laxenaire et al., 2018).

The North Pacific, with its complex circulation structure, is a region of highly active mesoscale eddy activity. It hosts intricate current systems such as the North Equatorial Current, the Kuroshio, and the North Pacific Current, where interactions between different flow systems generate a large number of mesoscale eddies. Previous studies have identified two zonal bands with high eddy kinetic energy in this region: one is the Kuroshio Extension, and the other is the North Pacific Subtropical Countercurrent region (Kang et al., 2010; Chang and Oey, 2014). In the Kuroshio Extension, eddy activity levels are modulated by the dynamical state of the Kuroshio and influenced by the wind stress curl in the eastern North Pacific. Both the Kuroshio and regional eddy intensity exhibit significant



decadal oscillations (Qiu and Chen, 2005; Qiu and Chen, 2013; Taguchi et al., 2010). The northern part of the Kuroshio Extension is dominated by anticyclonic eddies, while the southern part is predominantly cyclonic (Itoh and Yasuda, 2010). The North Pacific Subtropical Countercurrent region features complex circulation due to vertical shear with the North Equatorial Current, leading to notable eddy activity. Previous studies have completed eddy identification (Hwang et al., 2004) and dataset construction (Liu et al., 2012) in this area, confirming that eddies intensify as they propagate westward and influence the Kuroshio. Eddy kinetic energy in this region also exhibits an annual cycle (Qiu et al., 2010, 2013; Chow et al., 2017), with baroclinic instability being a key driver of its seasonal variability (Chang and Oey, 2014). Additionally, the spatiotemporal variability of eddies is regulated by baroclinic instability (Travis et al., 2017; Rieck et al., 2018). The waters around the Hawaiian Islands are a high-occurrence zone for eddies (Calil et al., 2008), where leeward eddy kinetic energy shows significant periods of approximately 60 and 100 days, as well as pronounced seasonal and interannual variations (Yoshida et al., 2011). The spatiotemporal resolution of northeasterly trade winds and regional wind forcing significantly influences the distribution of eddy kinetic energy in this area (Calil et al., 2008). In the California Current System, eddies frequently form near capes, islands, and regions with sharp topographic variations (Dong et al., 2012; Kurczyn et al., 2012).

Satellite altimetry has played an indispensable role in research areas such as the observation of mesoscale oceanic eddies (Chelton et al., 2007; Andres et al., 2008;). Studies based on the geostrophic (GEO) balance assumption—using satellite altimetry datasets such as AVISO/CMEMS—have achieved considerable progress in eddy identification, tracking, and the analysis of certain dynamical features. However, because classical geostrophic theory neglects the contribution of centrifugal forces in actual ocean flow, altimeter-derived sea surface velocities often exhibit biases in curved flow regions under the geostrophic balance assumption (Uchida et al., 1998; Fratantoni, 2001; Uchida and Imawaki, 2003; Douglass and Richman, 2015; Cao et al., 2023). These biases are distinct from the ageostrophic velocity components induced by surface wind stress. Corrections to altimeter-derived flow fields based on cyclogeostrophic balance theory have demonstrated superior performance in characterizing eddy dynamics in certain regional seas, such as the Gulf Stream, the Kuroshio Extension, the intense eddy zone in the Mozambique Channel, and the Mediterranean Sea (Fratantoni, 2001; Uchida and Imawaki, 2003; Penven et al., 2014; Ioannou et al., 2019). Although a wealth of findings has been accumulated in previous studies, little has been reported regarding practical implementation in theoretical

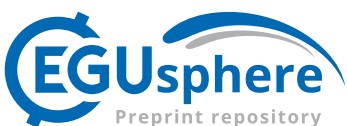

frameworks. In terms of theoretical application, the limitations of geostrophic balance theory lead to distorted representations of eddy dynamic characteristics in the complex and variable flow environment of the North Pacific. Meanwhile, systematic studies on the application of cyclogeostrophic balance theory in this region remain scarce, particularly in comprehensively comparing different types of eddies under both balance regimes. Using the global surface cyclogeostrophic current dataset corrected by Cao et al. 2023 based on the latest version of AVISO satellite altimetry data, this study compares eddy characteristics in the North Pacific between this refined dataset and the original altimetry product. Through quantitative diagnostic analysis, we aim to clarify the mechanism by which curvature influences the dynamic processes of surface flow fields.

## 2 Materials and Methods

### 2.1 Remote sensing data

The geostrophic current velocity fields were derived from the AVISO multi-satellite altimeter data product, distributed by the French Space Agency. This product synthesizes data from multiple satellite missions, including TOPEX/Poseidon, Jason-1, ERS, Envisat, and others, processed through the Data Unification and Altimeter Combination System (DUACS). This study utilizes the AVISO DT 2018 dataset, which has a daily temporal resolution and a spatial resolution of 0.25° × 0.25°, covering the period from 1 January 1993 to 31 December 2018. The data are accessible at: https://cds.climate.copernicus.eu/datasets/satellite-sea-level-global?tab=overview.

The cyclogeostrophic current data employed in this study were derived from the geostrophic currents calibrated by Cao et al. (2023). This dataset accounts for the influence of eddy curvature and was generated by applying an iterative method to perform cyclogeostrophic correction on 26 years of global surface geostrophic currents from the AVISO product.

### 2.2 Kinematic and dynamic parameters

Eddy Kinetic Energy (EKE) is a key measure of oceanic mesoscale activity(Quattrocchi et al., 2021). It is derived from geostrophic velocity anomalies and is defined as:

$$EKE = \frac{1}{2} \times \left[ \left( u' \right)^2 + \left( v' \right)^2 \right] \tag{1}$$

where $u'$ and $v'$ are the surface velocity anomalies of zonal and meridional currents.



The normalized difference of the mean EKE is defined as:

$$a_{EKE} = \frac{\overline{EKE_i} - \overline{EKE_g}}{\overline{EKE_g}} = \frac{\sum_{i=1}^{n} \frac{1}{2}\left(\left(u_i'\right)^2 + \left(v_i'\right)^2\right) - \sum_{i=1}^{n} \frac{1}{2}\left(\left(u_g'\right)^2 + \left(v_g'\right)^2\right)}{\sum_{i=1}^{n} \frac{1}{2}\left(\left(u_g'\right)^2 + \left(v_g'\right)^2\right)} \tag{2}$$

where, $a_{EKE}$ is the percentage difference in eddy kinetic energy, $EKE_i$ denotes the eddy kinetic energy of the cyclogeostrophic flow, $EKE_g$ represents the eddy kinetic energy of the geostrophic flow. $u_i'$, $v_i'$ are the zonal and meridional velocity anomalies of the cyclogeostrophic flow, respectively, while $u_g'$ and $v_g'$ denote the corresponding velocity anomalies of the geostrophic flow.

Enstrophy is a quantity directly related to kinetic energy in flow models that account for dissipative effects. On each grid cell, the eddy enstrophy is given by:

$$Ens = \frac{1}{2} \times \left(\frac{\partial v}{\partial x} - \frac{\partial u}{\partial y}\right)^2 \Delta x \Delta y \tag{3}$$

where $\Delta x$ and $\Delta y$ represent the zonal and meridional lengths of the grid point.

The strain rate S is composed of both shear $\vartheta = \frac{\partial v}{\partial x} + \frac{\partial u}{\partial y}$ and stretching deformation $\varsigma = \frac{\partial u}{\partial x} - \frac{\partial v}{\partial y}$ components, and is given by:

$$S = \sqrt{\vartheta^2 + \varsigma^2} \tag{4}$$

The normalized relative eddy vorticity, which quantifies the rotational characteristics of an eddy, is defined as:

$$\frac{\zeta}{f} = \frac{\partial v'/\partial x - \partial u'/\partial y}{f} \sim Ro \tag{5}$$

where $\zeta$ is the relative vorticity.

**2.3 Eddy Automated Detection Algorithm**

Eddies are defined as coherent fluid regions in which velocity vectors rotate cyclonically or anticyclonically about a central point. Automated eddy detection methods can be broadly categorized into Eulerian and Lagrangian approaches, depending on the type of underlying data (Nencioli et al., 2010). Within the Eulerian framework, eddies are typically identified from two- or three-dimensional flow field representations. In contrast, the Lagrangian approach detects eddies by tracking fluid particle



trajectories. In this study, a Eulerian eddy-detection algorithm is employed, which has been successfully applied to eddy identification in global oceans (Dong et al., 2022). The center of an eddy is determined based on the geometry of velocity vectors through four constraints:

1. Along the east-west direction through the eddy center, the surface velocity anomalies of the
meridional flow exhibit opposite signs on either side of the center and increase gradually away from it;

2. Along the north-south direction through the eddy center, the surface velocity anomalies of the zonal flow show opposite signs on either side of the center and increase gradually away from it;

3. The velocity magnitude reaches a local minimum in the vicinity of the eddy center;

4. Around the eddy center, the rotation of the velocity vectors is consistent. The directions of two
adjacent velocity vectors must lie within the same or two adjacent quadrants.

The minimum velocity detected by the first three constraints is not necessarily related to the eddy structure and cannot be directly defined as the eddy center. Therefore, the fourth constraint must be introduced to prevent false detection.

### 3 Results

**3.1 Eddy Kinetic Energy (EKE)**

Eddy Kinetic Energy (EKE) is an important parameter for characterizing eddy activity and serves as a key indicator for understanding the multi-scale dynamics of eddies. Cao et al. (2023) demonstrated that EKE after cyclogeostrophic adjustment exhibits marked differences across three distinct marine environments: low-latitude regions, eddy-rich zones with intense eddy-eddy interactions, and strongly
curved frontal systems. Figure 1a depicts the spatial distribution of the normalized difference between the multi-year mean geostrophic and cyclogeostrophic EKE in the North Pacific from January 1993 to December 2018. Based on this spatial pattern, five regions exhibiting the most pronounced differences were selected for detailed analysis: the Aleutian Islands (AI), the Kuroshio Extension (KE), the South China Sea (SCS), the California Coastal Current (CC) region, and the Hawaiian Islands (HI). The
precise geographical boundaries of these regions are provided in Table 1.

The most pronounced differences occur in the KE andSCS regions. In the KE region, positive anomalies are located north of the Kuroshio Extension axis, whereas negative anomalies are observed to the south. In the SCS region, factors such as complex topography and Kuroshio intrusions contribute



to abundant eddy activity(Lin et al., 2015; Zhang et al., 2016). However, satellite altimetry data tend to underestimate the intensity of strongly curved flows in this area. Driven by the combined effects of the Coriolis force and prevailing wind belts, the North Pacific Subtropical Gyre exhibits cyclonic circulation. Along its southern boundary, where currents flow southward, the EKE difference ratio is negative, notably reaching approximately -5% in the CC region. In contrast, the Subpolar Gyre demonstrates anticyclonic circulation. Influenced by the Alaska Current, the AI region shows an EKE

difference percentage around 10%. In the HI region, due to wind stress curl and current shear in the eastern area, positive differences are distributed in the west while negative differences appear in the east, both with magnitudes around ±5%.

Table 1 Location information of the selected research area

| Region | Latitude | Longitude |
|--------|----------|-----------|
| AI | 165°E-155°W | 48°N-56°N |
| KE | 140°E-175°W | 28°N-42°N |
| CC | 132°W-116°W | 20°N-40°N |
| SCS | 105°E-121°E | 5°N-25°N |
| HI | 168°E-154°W | 16°N-23°N |

This study compares the characteristics of EKE variability computed under two balance conditions across the study regions, and analysis interannual time series of area-averaged EKE for both cyclonic (positive relative vorticity) and anticyclonic (negative relative vorticity) eddies within each region. Figure 1b illustrates the interannual variability of EKE in the South China Sea (SCS) region. Satellite altimetry overestimates EKE for cyclonic eddies and underestimates it for anticyclonic eddies in this

area. The geostrophic EKE of cyclonic eddies exceeds the cyclogeostrophic EKE, with an annual mean difference of -9.69%. In contrast, the geostrophic EKE of anticyclonic eddies is lower than the cyclogeostrophic EKE, showing an annual mean difference of 12.28%. The maximum difference for anticyclonic eddies reached 16.95% in 2009, while the minimum was 5.46% in 2008. For cyclonic eddies, the maximum was -13.47% in 2008 and the minimum was -7.81% in 1998. Temporally, the

cyclogeostrophic correction proves more pronounced for anticyclonic eddies. Spatially, the difference percentages in this region are predominantly positive. In the CC region (Fig. 1c), the average geostrophic EKE for cyclonic and anticyclonic eddies is 80.4 cm²/s² and 70.2 cm²/s², respectively, while the average cyclogeostrophic EKE values are 64.7 cm²/s² and 70.9 cm²/s². The geostrophic EKE of cyclonic eddies is higher than the cyclogeostrophic EKE, with an annual mean difference of



-19.48%. In contrast, the geostrophic EKE of anticyclonic eddies is slightly lower than the

cyclogeostrophic value, showing an annual mean difference of 1.01%. Thus, the discrepancy between

geostrophic and cyclogeostrophic EKE is substantial for cyclonic eddies but minimal for anticyclonic

eddies in this region. These results highlight a pronounced divergence between the two EKE estimates

for cyclonic eddies in the CC region (Fig. 1a and c). In the KE region (Fig. 1d), the geostrophic EKE of

cyclonic eddies is higher than the cyclogeostrophic EKE, whereas the opposite is true for anticyclonic

eddies. The annual mean difference is -11.47% for cyclonic eddies and 7.21% for anticyclonic eddies.

The maximum difference for anticyclonic eddies is 10.25% (2004), and the minimum is 5.35% (1997).

For cyclonic eddies, the maximum difference is -13.45% (2005), and the minimum is -8.89% (2004).

Cyclogeostrophic correction is significant for both eddy types in this region. In the AI region (Fig. 1e),

differences in EKE between cyclonic and anticyclonic eddies under geostrophic and cyclogeostrophic

balances are relatively small. The geostrophic EKE of cyclonic eddies exceeds the cyclogeostrophic

EKE, with an annual mean difference of -0.93%. For anticyclonic eddies, the geostrophic EKE is lower,

yielding an annual mean difference of 5.10%. Cyclogeostrophic correction is more significant for

anticyclonic than cyclonic eddies. The maximum and minimum difference percentages for anticyclonic

eddies are 6.50% and 3.74%, respectively; for cyclonic eddies, they are -2.02% and 0.13%. Figure 1f

presents the relative differences between cyclogeostrophic and geostrophic EKE over time in the HI

region. The geostrophic EKE of cyclonic eddies is higher than the cyclogeostrophic EKE, while the

reverse occurs for anticyclonic eddies. The mean difference is -9.17% for cyclonic and 8.86% for

anticyclonic eddies. The extreme values are 11.25% and 5.65% for anticyclonic eddies, and -12.20%

and -6.32% for cyclonic eddies. Overall, the differences in EKE between eddies with negative and

positive relative vorticity show consistently opposite trends: cyclogeostrophic EKE is higher than

geostrophic EKE for cyclonic eddies, while cyclogeostrophic EKE is lower than geostrophic EKE for

anticyclonic eddies.

To quantify how curvature affects EKE, we normalized the average EKE and eddy lifetimes within

eddy boundaries for two datasets in the study area. By applying this normalization, we can track how

EKE changes over the eddy's full lifetime, as shown in Fig 2. Across the five regions, the average

normalized EKE exhibits a concave trend over the normalized eddy lifespan, first increasing and then

decreasing. This reflects the typical evolutionary pattern of energy during the development, maturity,



and decay stages of eddies. However, differences exist among various regions and eddy types in terms

of the amplitude of changes and the position of peaks.

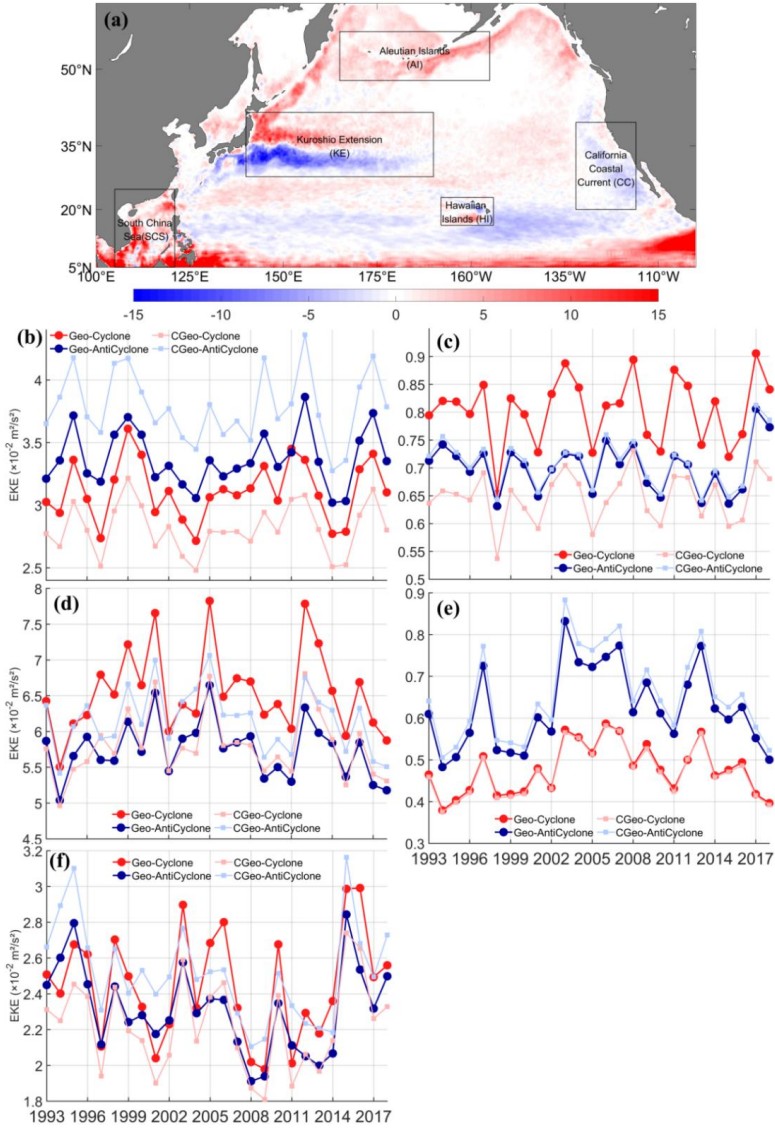

**Figure 1 (a) Spatial distribution of normalized difference between multi-year average geostrophic EKE and cyclogeostrophic EKE over the period of January 1993 to December 2018 in the North Pacific. Time series of interannual variation of area averaged EKE in the (b) South China Sea, (c) California Coastal Current region, (d) Kuroshio Extension, (e) Aleutian Islands, (f) Hawaii Islands, the dotted solid blue and red lines**
**represent geostrophic cyclonic and anticyclonic currents, respectively. The dotted solid light blue and red lines represent the cyclogeostrophic cyclonic and anticyclonic currents, respectively.**





In the SCS region (Fig. 2a), the normalized cyclogeostrophic EKE of cyclonic eddies consistently exceeds the geostrophic EKE throughout their entire lifecycle. During the generation phase, the growth rate of cyclogeostrophic EKE is lower than that of geostrophic EKE. In contrast, the intensification phase exhibits a more rapid increase in cyclogeostrophic EKE compared to geostrophic EKE. Throughout the mature phase, EKE values under both balance regimes converge and remain close, albeit with minor fluctuations. During the decay phase, the decline in cyclogeostrophic EKE is more gradual than that of geostrophic EKE. For anticyclonic eddies, the variation in cyclogeostrophic EKE is smaller than that of geostrophic EKE during both the generation and decay phases. In the intensification and mature phases, the normalized cyclogeostrophic EKE remains lower than the geostrophic EKE. Although the geostrophic EKE accelerates slightly faster during the generation phase, the overall difference is modest. Notably, during the decay phase, cyclogeostrophic EKE is retained more effectively, resulting in a significantly slower decay rate compared to geostrophic EKE. Figure 2b displays the evolution of average EKE within eddy boundaries in the CC region over the full eddy life cycle. Throughout the generation and decay phases, the normalized cyclogeostrophic EKE exhibits less variation than the geostrophic EKE. Under cyclogeostrophic balance, the normalized EKE values of cyclonic and anticyclonic eddies are similar during initial generation and final decay, with the difference between the two eddy types being markedly smaller than under geostrophic balance. Furthermore, the decay rate of cyclogeostrophic EKE is significantly slower than that of geostrophic EKE. Anticyclonic eddies are also less influenced by the western boundary current, resulting in comparatively weaker modulation of their EKE differences relative to cyclonic eddies. In the AI region (Fig. 2c), the normalized EKE of both cyclonic and anticyclonic eddies exhibits similar peak values and evolutionary trends under geostrophic and cyclogeostrophic balance, with discernible disparities emerging primarily around the termination phase. During the generation phase, the energy acceleration rates of geostrophic and cyclogeostrophic EKE are comparable for cyclonic eddies, whereas anticyclonic eddies show a weaker increase in cyclogeostrophic EKE than in geostrophic EKE. In the decay phase, the cyclogeostrophic EKE declines more rapidly than the geostrophic EKE. In the Kuroshio Extension (KE) region (Fig. 2d), cyclonic eddies exhibit similar normalized EKE values under both balance conditions. In contrast, for anticyclonic eddies, the difference in EKE between the two balances initially widens before narrowing during the generation and development phases, a pattern that persists through the mature and decay stages. During the generation phase, the rate of



increase in cyclogeostrophic EKE is lower than that of geostrophic EKE for anticyclonic eddies.

Conversely, in the decay phase, cyclogeostrophic EKE demonstrates significantly higher persistence,

declining at a much slower rate compared to geostrophic EKE. Figure 2e illustrates that in the HI

region, cyclonic eddies exhibit higher normalized cyclogeostrophic EKE than anticyclonic eddies

during the generation phase. However, the growth rate of normalized cyclogeostrophic EKE is lower

than that of geostrophic EKE for both eddy types. Throughout the mature stage and early decay phase,

cyclogeostrophic EKE shows a more rapid decline compared to geostrophic EKE.

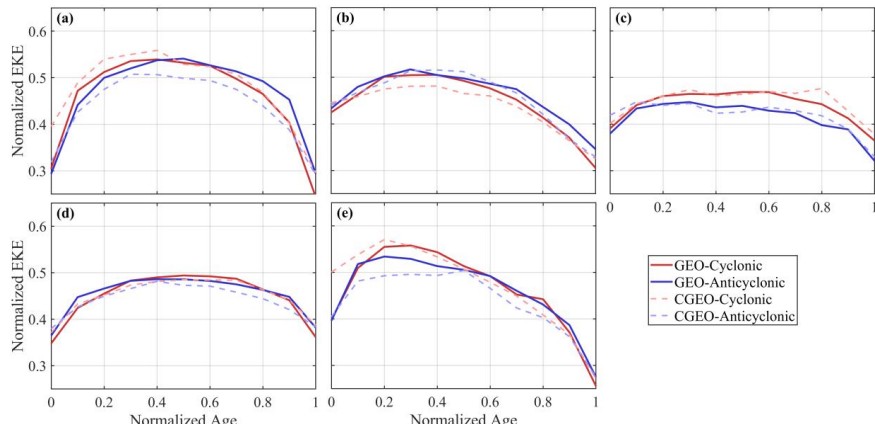


**Figure 2 Temporal evolution of normalized eddy kinetic energy in (a) South China Sea (SCS), (b) California Current (CC), (c) Agulhas Insurgence (AI), (d) Kuroshio Extension (KE), and (e) Hawaiian Islands (HI). Analysis restricted to eddies persisting >28 days. Solid red line represents geostrophic cyclonic eddies and dashed red line represents cyclogeostrophic cyclonic eddies. The solid and dashed blue lines indicate**
**geostrophic and cyclogeostrophic anticyclonic eddies, respectively. Each life stage is defined by normalized age: generation (~0-0.1), intensification (~0.1-0.3), maturation (~0.3-0.8), and decay (>0.8).**

**3.2 Eddy Spatiotemporal Distributions, Sizes and Lifespan**

An analysis and comparison of the number, radius, and lifespan of eddies (with lifespans exceeding 4

weeks) were carried out across five study regions in the North Pacific between 1993 and 2018. The

Lagrangian method was applied to quantify eddy counts by treating each complete eddy lifecycle as a

single unit. In terms of eddy numbers (Fig. 3a), the total under cyclogeostrophic balance is generally

lower than that under geostrophic balance. Specifically, the geostrophic balance yields approximately

35.65% more eddies than the cyclogeostrophic balance.In the AI and KE regions, anticyclonic eddies

predominate over cyclonic eddies, while in the CC and HI regions, cyclonic eddies are more prevalent

than anticyclonic eddies. To gain a clearer understanding of the distribution of eddies in different



regions, a 1° × 1° grid was used to delineate the statistical areas. The counted eddies were confined to those with their entire lifecycle (from generation to dissipation) within the study area. Based on the center position of each eddy at each moment, all eddies within each grid point were counted . Overall, fewer eddies were detected under cyclogeostrophic conditions (Figure 3d–e) than under geostrophic

conditions (Figure 3b–c). Despite this difference, both frameworks show strong spatial correspondence across all regions. In the AI region, eddy occurrence is relatively sparse near the central coastal margin, whereas higher concentrations are observed in the northwestern and southeastern areas. The KE region exhibits a substantial number of eddies under both geostrophic and cyclogeostrophic conditions, with cyclonic eddies dominating the southern and eastern zones, and anticyclonic eddies more frequent in

the northern and southwestern sectors. West of the Hawaiian Islands in the HI region, a localized area shows a notable density of eddies in both GEO and CGEO datasets. In the CC region, eddy generation is enhanced along the northeastern coastline due to coastal upwelling. By contrast, eddy activity is relatively low in coastal zones and near the equator.

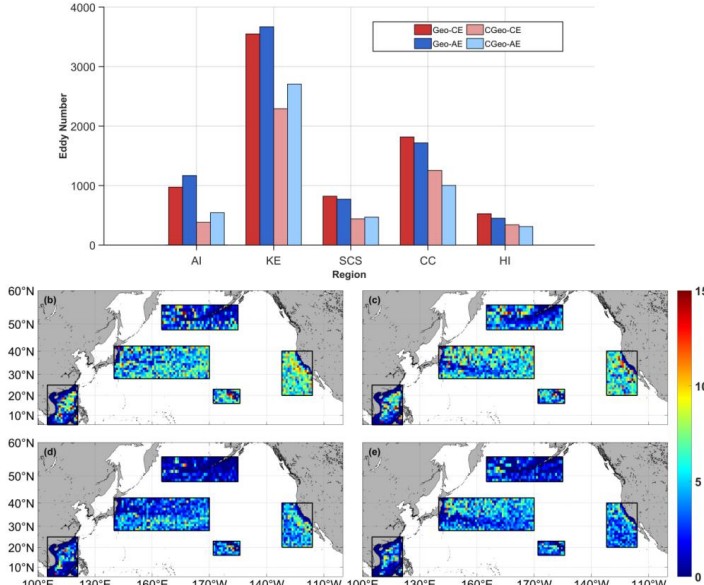

**Figure 3 Comparison of eddy numbers. (a) Histograms of eddy numbers across the five study regions in the North Pacific. Geo-CE and Geo-AE denote geostrophic cyclonic and anticyclonic eddies, respectively; CGeo-CE and CGeo-AE indicate cyclogeostrophic cyclonic and anticyclonic eddies. (b–c) Spatial distributions of GEO-CE and GEO-AE. (d–e) Spatial distributions of CGEO-CE and CGEO-AE within the North Pacific study regions.**



Regarding the average lifespan of eddies, the number of eddies with a lifespan of at least 4 weeks is shown in Fig. 4a. The average eddy lifespan under cyclogeostrophic conditions is shorter than under geostrophic conditions, with a difference of approximately 10 days in the Kuroshio Extension (KE) and California Current (CC) regions. The average lifespans under the two balance conditions are most similar in the South China Sea (SCS). Except in the CC region, the average lifespan of anticyclonic

eddies is longer than that of cyclonic eddies in the other four regions. As for the average eddy radius (Fig. 4b), cyclogeostrophic eddies are generally larger than geostrophic eddies. A notable difference in average radius is observed between anticyclonic eddies (AE) and cyclonic eddies (CE) — under cyclogeostrophic conditions, both types exhibit a larger average radius compared to those under geostrophic conditions.


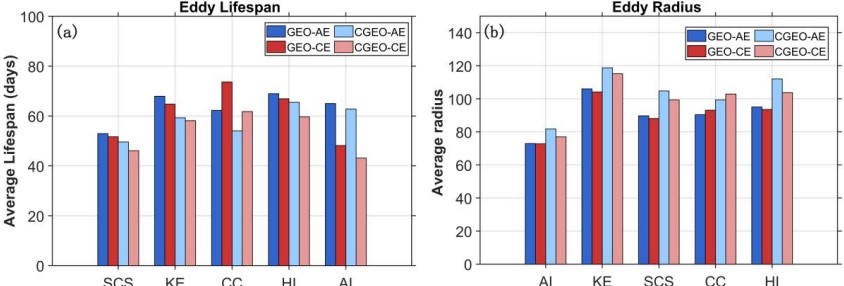

**Figure 4 Statistics of (a) eddy lifespans (lifetime $\geq$ 4 weeks) and (b) radii across the five study regions in the North Pacific.**

Figure 5 displays the distribution of eddy radii, showing a consistent right-skew across both datasets

from the five regions. Eddy radii derived under the CGEO framework are generally larger than those under the GEO framework. In the CGEO dataset, cyclonic eddies (CE) with radii between 50–100 km comprise 57.5% of the total, and anticyclonic eddies (AE) account for 56%. In contrast, the GEO dataset shows higher proportions in this size range, with 65% for CE and 63.2% for AE, indicating a greater abundance of small-radius eddies.




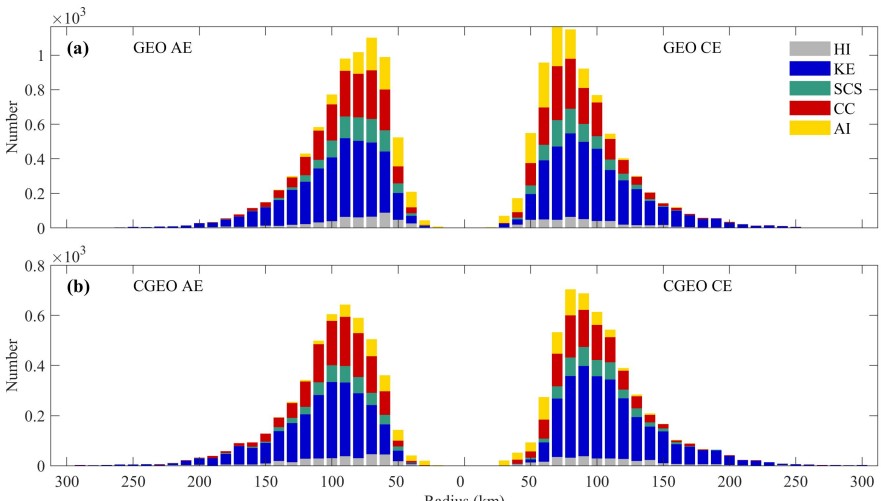

**Figure 5 Histograms of eddy radius for the (a) GEO and (b) CGEO datasets.**

Statistical analysis of eddy lifespans (Figure 6) reveals that short-lived eddies are more numerous than long-lived eddies in both datasets. As lifespans increase, the numbers of both anticyclonic eddies (AEs) and cyclonic eddies (CEs) decrease. This occurs because the energy sustaining an eddy continuously dissipates into the surrounding environment after its formation. Consequently, eddies with longer lifespans become increasingly difficult to maintain, ultimately leading to their dissipation. Figure 6 shows that eddies in the CGEO dataset generally have shorter lifespans than those in the GEO dataset. In the GEO dataset (Figure 6a), eddies with lifespans of 4–10 weeks account for 77.5% of AEs and 77.3% of CEs. In contrast, the CGEO dataset shows higher proportions in this lifespan range: 82.9% for AEs and 80.9% for CEs (Figure 6b). A notable difference between the two datasets is observed for eddies lasting longer than 20 weeks. Only 5.3% of AEs and 5.2% of CEs in the GEO dataset fall into this category, compared to 3.6% of AEs and 4.2% of CEs in the CGEO dataset. Among all eddies with lifespans exceeding 10 weeks, the GEO dataset contains 22.2% AEs and 22.7% CEs, whereas the CGEO dataset contains 17.1% AEs and 19.1% CEs. These results indicate that CEs generally have longer lifespans than AEs.



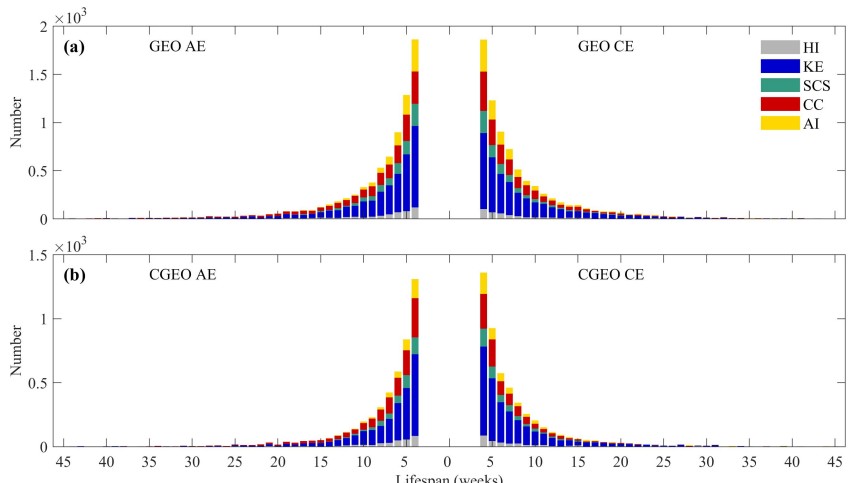

**Figure 6 Histograms of eddy lifetimes, with (a) for GEO and (b) for CGEO.**

To investigate the spatial variation of eddy radius, an Eulerian approach was employed to classify eddy
        radii. Consistent with previous studies, the research area was divided into 1°×1° bins, where the value
        of each bin represents the average eddy radius of the corresponding pixels. Overall, eddies near the
        coastline tend to have a smaller radius, while the radius gradually increases with increasing distance
        from land (Fig. 7). The corrected cyclogeostrophic eddy(Fig. 7c and d) radius data are more affected by

land, resulting in more blank coastal areas compared to the geostrophic eddy(Fig. 7a and b) radius data.
        Additionally, eddies under cyclogeostrophic conditions have a larger radius than those under
        geostrophic conditions, and the eddy radius distribution patterns of the two datasets remain highly
        similar.In the KE and SCS regions, eddies in the central areas have larger radii, while those at the edges
        have smaller radii. In the CC and HI regions, eddy radius is small along the northeastern coast and

increases toward the southwest. In the AI region, larger eddy radii are observed in the southwest, while
        smaller radii are found along the northeastern coast. In parts of the KE and HI regions, the maximum
        eddy radius reaches up to 140 km.In the KE region, the large curvature of the Kuroshio Extension's
        main axis causes instability, which often leads to the detachment of eddies with larger radii and longer
        lifespans from the jet stream. In the southwest of the HI region, multiple factors including

topographically forced trailing effects, Ekman pumping from wind stress curl, and nonlinear
        background circulation modulation collectively lead to the formation of eddies with larger radii.



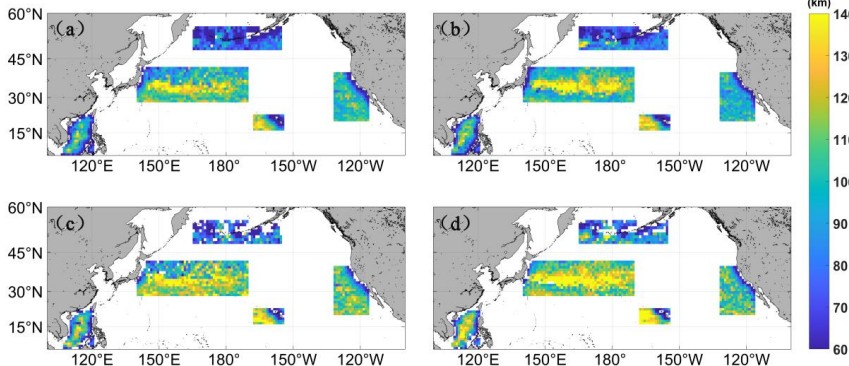

**Figure 7 Spatial distributions of eddy radius for (a) CEs and (b) AEs identified under geostrophic balance conditions, and for (c) CEs and (d) AEs identified under cyclogeostrophic balance conditions.**

365 The average normalized eddy radius across the five regions exhibits a similar concave-up trend with the normalized eddy lifespan parameters and eddy kinetic energy (EKE) (Fig. 8). The eddy radius of both CGEO and GEO shows a rapid increase during the formation phase, a slight growth slowdown during the development phase, stability during the mature phase, and a rapid decline during the dissipation phase.In the AI region (Fig. 8a), the normalized eddy radius curves of anticyclonic eddies

370 are similar between the GEO and CGEO datasets during the eddy formation phase. For cyclonic eddies, however, the normalized radius of CGEO is significantly larger than that of GEO in the early formation stage. After correction of the CGEO data, the growth rate of CGEO eddies during the formation stage is lower than that of GEO eddies; during the mature phase, the normalized radius of CGEO eddies fluctuates more in both amplitude and frequency compared to GEO eddies. In the dissipation phase, the

375 radius decay rate of GEO eddies is faster than that of CGEO eddies.In the KE region (Fig. 8b), the growth rate of CGEO eddies during the generation phase is lower than that of GEO eddies, while the radius decay rate of GEO eddies during the decay phase is higher than that of CGEO eddies. At both the initial generation and final decay stages, the radius of CGEO eddies is greater than that of GEO eddies; however, during the maturation phase, the maximum radius of CGEO eddies is smaller than

380 that of GEO eddies.In the SCS region (Fig. 8c), the growth rate of CGEO eddies during the generation phase and the radius decay rate during the decay phase are both lower than those of GEO eddies. The radius of CGEO eddies is larger than that of GEO eddies at the initial generation and final decay stages. In the maturation phase, the peak radius of GEO-CE is slightly higher than that of CGEO-CE, while the peak radii of GEO-AE and CGEO-AE are similar.In the CC region (Fig. 8d), the growth rate of CGEO



eddies during the generation phase and the radius decay rate during the decay phase are both lower than those of GEO eddies. At the initial generation stage, the radius of CGEO-AE is greater than that of GEO-AE, while the radii of CEs under the two balance conditions are similar. At the final decay stage, the radius of CGEO-CE is greater than that of GEO-CE, while the radii of AEs under the two balance conditions are similar. Additionally, during the generation phase, the growth rate of CGEO-CE radius

is greater than that of GEO-CE, while the growth rates of CGEO-AE and GEO-AE are similar; the opposite trend is observed during the decay phase.n the HI region (Fig. 8e), the growth rate of CGEO eddies during the generation phase and the decay rate during the decay phase are both lower than those of GEO eddies. When eddies are initially generated, the radius of CGEO-CE is larger than that of GEO-CE, while the radii of AE under the two balance conditions are similar. At the final decay stage,

the radius of CGEO eddies remains greater than that of GEO eddies.Overall, across the five regions, the radii of GEO eddies are comparable between the initial generation and final decay phases. For the modified CGEO dataset, the growth of AE during the generation phase is greater than that of CE, while the decay of AE during the decay phase is slower than that of CE. When CGEO eddies are initially generated, the radius of CE is larger than that of AE; however, by the final decay phase, the radius of

CE becomes smaller than that of AE.Across the five regions, the growth rate of CGEO eddies during the generation phase is lower than that of GEO eddies, while during the decay phase, the radius decay rate of GEO eddies is faster than that of CGEO eddies. Additionally, AE in the modified CGEO dataset exhibits better radius retention characteristics.



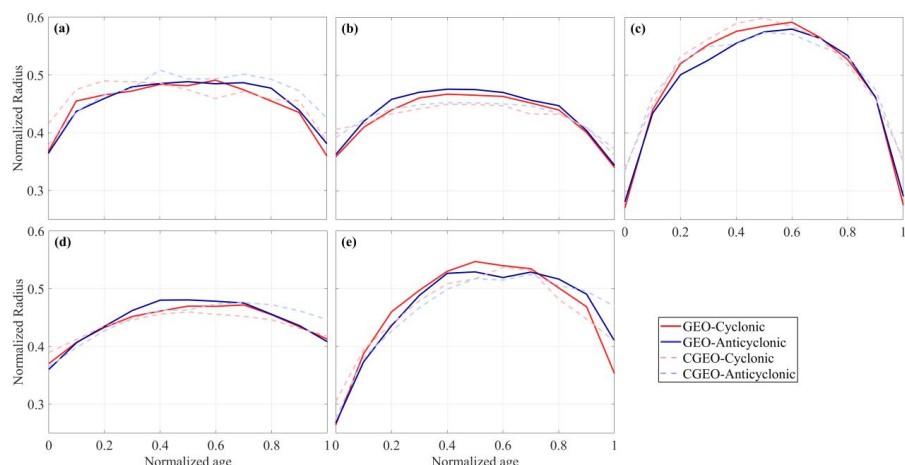


**Figure 8 Temporal evolution of the mean normalized radius for the (a) SCS, (b) CC, (c) AI, (d) KE, and (e) HI regions.**

### 3.3 Eddy Generation and Dissipation

Figure 9 displays the interannual average time series of eddy generation numbers across the five study

areas from January 1993 to December 2018. Overall, the GEO dataset exhibits a higher number of

eddy generation events than the CGEO dataset, though both share similar interannual variability. In the

AI region (Fig. 9a), CGEO-AE and CGEO-CE average 15.1 and 21.8 eddies per year, respectively,

considerably fewer than the GEO averages of 36.7 (AE) and 44.0 (CE), corresponding to mean

percentage differences of 58.9% (AE) and 50.3% (CE). The interannual ranges are 30.7–74.2% for AE

and 29.7–66.0% for CE. As presented in Fig. 9b for the CC region, the mean eddy generation under

CGEO balance is 47.7 (AE) and 39.9 (CE), lower than the GEO averages of 66.6 (AE) and 64.2 (CE).

This corresponds to a mean percentage difference of 37.8% for AE and 28.5% for CE, with interannual

maxima reaching up to 49.1%. In the KE region (Fig. 9c), CGEO yields annual averages of 88.8 (AE)

and 103.9 (CE), notably lower than the GEO values of 131.2 (AE) and 135.0 (CE). The mean

percentage difference was significantly higher for CE (32.3%) than for AE (23.0%). Interannually, the

values for AE ranged from 22.5% in 2009 to 44.4% in 1995, whereas for CE, they ranged from 8.2% in

2014 to 32.6% in 2018. In the HI region (Fig. 9d), CGEO shows annual averages of 13.6 (AE) and 13.3

(CE), below the GEO values of 19.8 (AE) and 18.3 (CE). The mean percentage difference is 31.4% for

CE and 26.9% for AE. A notable feature here is the occurrence of negative differences: the minimum

for AE was -7.1% in 1994, with a maximum of 59.3% in 1997; for CE, the minimum was -16.7%





(2002) and the maximum was 50.0% (2012). In the SCS region (Fig. 9e), the annual average generation

numbers for CGEO are 16.8 (AE) and 18.1 (CE), compared to 28.8 (AE) and 27.4 (CE) for GEO. The

mean percentage difference is 41.8% for CE and 33.9% for AE. The maximum difference for AE was

60.6% in 2015, with a minimum of 15.4% in 1999; for CE, the values were 54.8% (2004) and 8.3%

430     (2007).

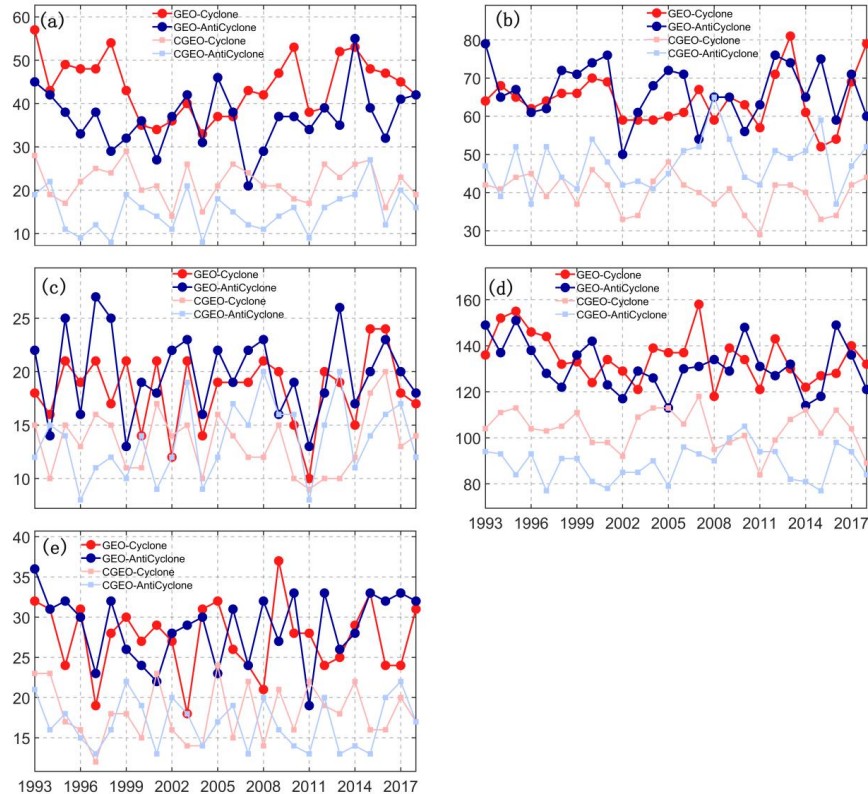

**Figure 9 Interannual average time series of eddy generation numbers from January 1993 to December 2018
for (a) AI, (b) CC, (c) KE, (d) HI, and (e) SCS. The dashed blue and red lines represent GEO-CE and
GEO-AE, respectively. The dashed solid light blue and red lines represent CGEO-CE and CGEO-AE,
respectively.**

Similar to the generation phase, the GEO dataset exhibits higher eddy dissipation numbers than the

CGEO dataset, with both following comparable interannual variation patterns. In the AI region (Fig.

10a), the annual average number of dissipations for CGEO-AE and CGEO-CE is 14.7 and 20.7,

respectively, considerably lower than the corresponding GEO values of 37.3 and 43.2. This results in a



percentage difference of 60.6% for AE and 52.2% for CE. The maximum difference reached 82.4% for AE in 1996 and 63.2% for CE in 2002, while the minimum values were 40.4% for AE in 2015 and 33.3% for CE in 2001. In the CC region (Fig. 10b), CGEO-AE and CGEO-CE have annual average dissipation numbers of 48.1 and 37.3, respectively, compared to 65.2 and 59.1 under GEO conditions.

The resulting percentage differences are 26.2% for AE and 36.9% for CE. The interannual extremes include a maximum difference of 44.4% for AE in 1994 and 55.8% for CE in 2011, and a minimum of 1.5% for AE in 2008 and 8.7% for CE in 2005. The KE region (Fig. 10c) shows annual average dissipation counts of 88.8 for CGEO-AE and 103.2 for CGEO-CE, against 129.5 and 132.4 for their GEO counterparts, leading to differences of 31.5% and 21.8%, respectively. The highest differences

were observed in 1995 for AE (39.5%) and in 1996 for CE (33.6%), whereas the lowest occurred in 2018 for AE (22.2%) and in 2016 for CE (10.4%). In the HI region (Fig. 10d), CGEO-AE and CGEO-CE dissipate at averages of 14.7 and 11.7 per year, respectively, versus 21.2 and 16.8 under GEO, both with a difference around 30%. Notably, the maximum difference peaked at 52% for AE in 1997 and 66.7% for CE in 1994, while the minimum was -5.6% for AE in 2009 and 5.3% for CE in

2018. Finally, in the SCS region (Fig. 10e), the annual average dissipation numbers for CGEO-AE and CGEO-CE are 17.0 and 18.1, respectively, lower than the GEO values of 30.0 and 28.2, corresponding to differences of 43.1% and 35.8%. The maximum difference for both types peaked in 1996, at 59.3% for AE and 55.2% for CE, while the minimum was recorded in 1999 for AE (10.7%) and in 2017 for CE (8.7%).




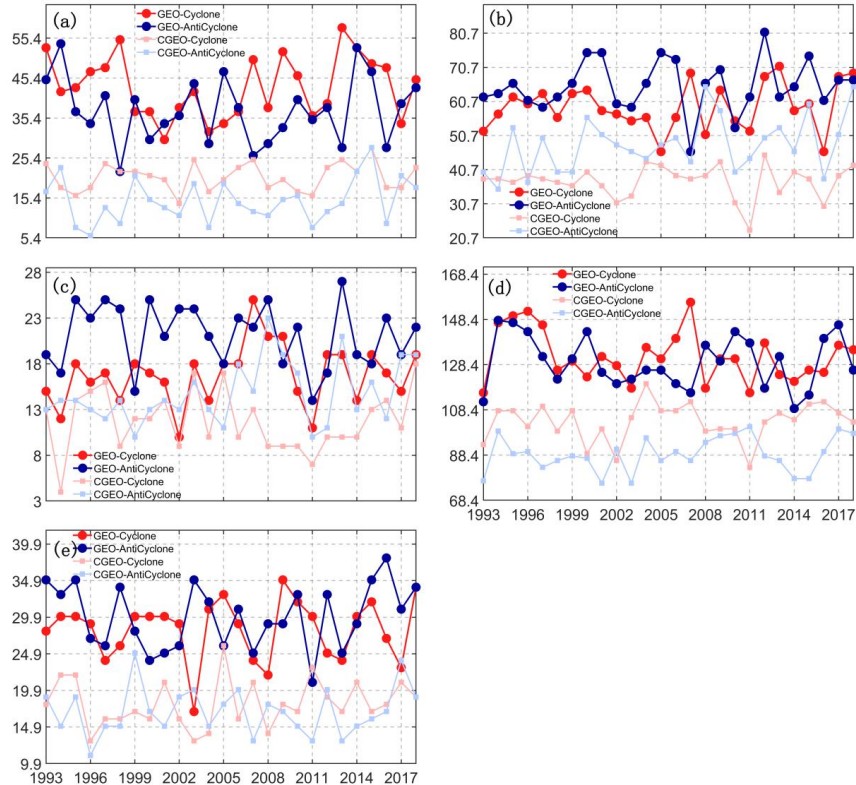

**Figure 10 Interannual average time series of eddy dissipation numbers from January 1993 to December 2018 for (a) AI, (b) CC, (c) KE, (d) HI, and (e) SCS. The dashed blue and red lines represent GEO-CE and GEO-AE, respectively. The dashed solid light blue and red lines represent CGEO-CE and CGEO-AE, respectively.**

The spatial distributions of eddy generation and dissipation are presented in Figures 11 and 12, respectively, with the analysis based on 1°×1° grids and including only eddies that persist for more than four weeks. Overall, the number of eddies generated and dissipated under cyclogeostrophic (CGEO) balance is lower than under geostrophic (GEO) balance; however, the spatial patterns of eddy distribution remain similar under both dynamical frameworks. In the AI region, GEO eddy abundance is higher in the northwestern and southeastern sectors, whereas CGEO eddies are more numerous in the eastern part of the region. Notably, the CE count surpasses that of AE in this region. In the KE region, the two datasets exhibit a notable consistency: fewer eddies are generated along the central axis of the Kuroshio Extension, while more AEs (to the north) and CEs (to the south) are produced. In the SCS region, eddy generation occurs more frequently in the eastern sector, including



the Luzon Strait and west of the Philippines. In addition, markedly fewer CGEO eddies(Fig. 11c and d) are observed in the southern SCS compared to GEO eddies(Fig. 11a and b). In the CC region, eddy generation is predominantly concentrated in the eastern and southwestern coastal zones. In the HI region, eddies are clustered to the west of the Hawaiian Islands, and the number of CGEO-AEs(Fig. 11c) exceeds that of CGEO-CEs(Fig. 11d).


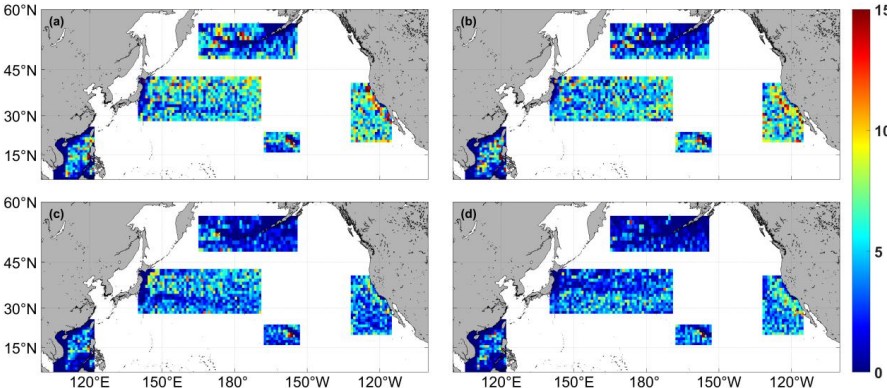

**Figure 11 Spatial distribution of the number of eddies generated. The AEs and CEs detected by geostrophic currents are shown in (a) and (b), respectively. The AEs and CEs detected by cyclogeostrophic currents are shown in (c) and (d).**


As shown in Figure 12, in the KE region, cyclonic eddies (CEs) (Fig. 11b and d)are predominantly distributed south of the Kuroshio Extension axis, while anticyclonic eddies (AEs)(Fig. 11a and c) are concentrated to the north. In the CC region, eddy dissipation occurs mainly in the coastal areas, whereas in the HI region, it is distributed along the western coast of the Hawaiian Islands. In the AI region, the dissipation of GEO eddies is largely concentrated in the eastern and southwestern sectors, while that of CGEO eddies is focused primarily in the east. In the SCS region, the number of CGEO eddies generated in the southern part is notably lower than the number of GEO eddies in other areas of the basin.




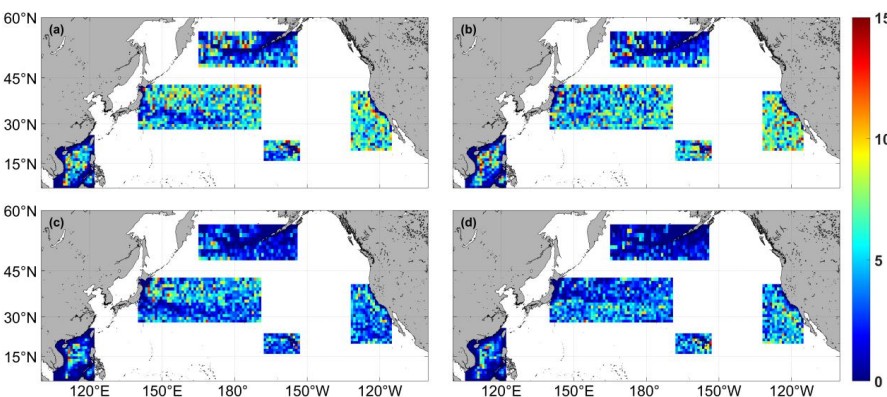


**Figure 12 Same as Figure 11, but for the number of eddy dissipations.**

### 3.4 Movement and Propagation

As revealed by the statistical analysis of zonal velocity in Fig.13, CEs and AEs generally exhibit similar translational speeds and directions in most regions, with the exception of certain areas within

the South China Sea (SCS). The average difference in eddy translational speed is mostly below 0.5 cm/s, with CGEO eddies displaying greater speed variability than GEO eddies. Eddies in all five regions propagate westward. However, in all regions except HI, the westward speed decreases as latitude increases. Regional differences are also observed in the meridional movement: only in the CC region do both CEs and AEs move southward, with CEs propagating more slowly than AEs in that area.

Further analysis of the relationship between westward speed and latitude indicates that in the AI and HI regions, the westward speed of CEs derived from CGEO data exceeds that from GEO data. Moreover, CGEO eddies exhibit more pronounced speed fluctuations than GEO eddies, generally following the pattern of increasing westward speed with decreasing latitude. The maximum westward speed occurs in the HI region around 20°N. In terms of northward velocity (Fig. 13b, d), CGEO also displays greater

variability than GEO. Overall, the SCS region shows the largest speed fluctuations among all regions. Additionally, variations in westward speed have a stronger influence on CEs (Fig. 13a) than on AEs (Fig. 13c).





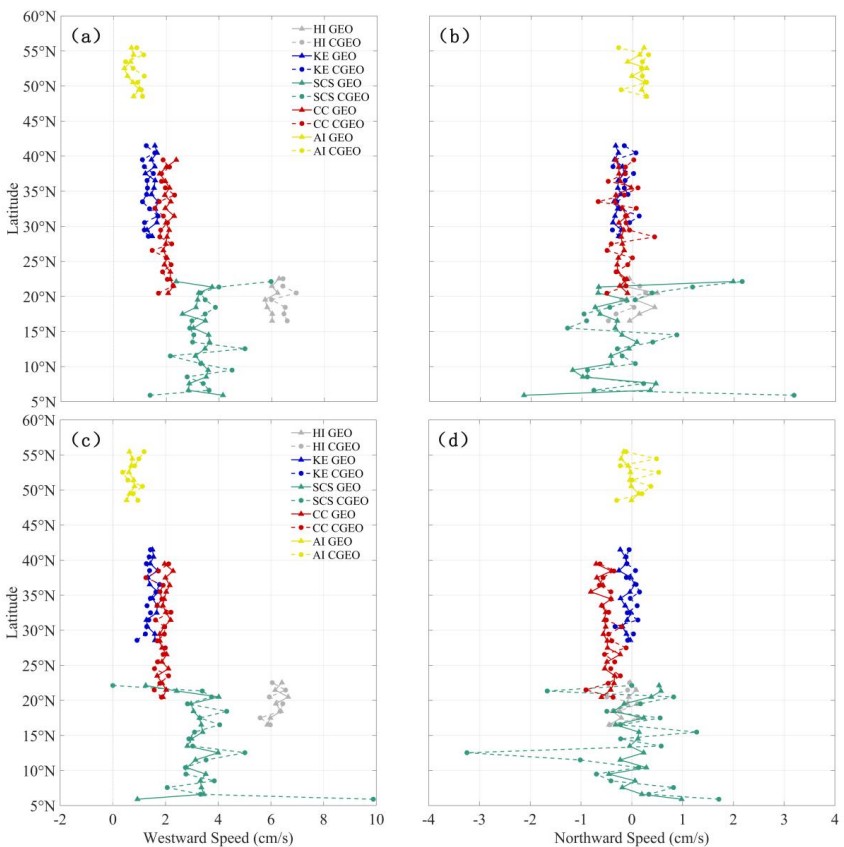

**Figure 13 Latitudinal variations in the (a) zonal and (b) meridional speeds of CEs, and (c) zonal and (d) meridional speeds of AEs. Dashed and solid lines denote results from the CGEO and GEO balances, respectively..**

According to the meridional statistics, the speed variation of CGEO is more fluctuating than that of GEO. Both CGEO and GEO eddies mainly move westward, with speeds ranging from 0 to 7 cm/s. In the HI region, the westward speed is faster, ranging from 4 to 14 cm/s, while AI, KE, and CC occasionally exhibit eastward fluctuations, with speeds under 3 cm/s. In the AI region, the CE (AE) moves eastward around 176°W (171°W) in CGEO data, with a speed of around 7 cm/s. The westward movement speed in the AI region around 165°W shows a large speed difference between the two datasets, with a difference greater than 5 cm/s. In the SCS, the speed difference between the two datasets is significant around 103°E, exceeding 7 cm/s. For CGEO-CE, the northward speed variation is large in the SCS region around 102°E, the western KE, and the eastern AI region, with northward




and southward speeds reaching about 5 cm/s. In other regions, the speed fluctuations remain within the range of -2 to 2 cm/s. For CGEO-AE, the northward speed variation is also large in the SCS region around 102°E, the western KE, and the eastern AI region, but the maximum values in the SCS and AI regions are higher than those for CE, while the maximum value in KE is smaller than that for CE. The data correction has a greater impact on AE than on CE, with the speed difference in the northward movement for AE being noticeably larger than that for CE in both datasets.

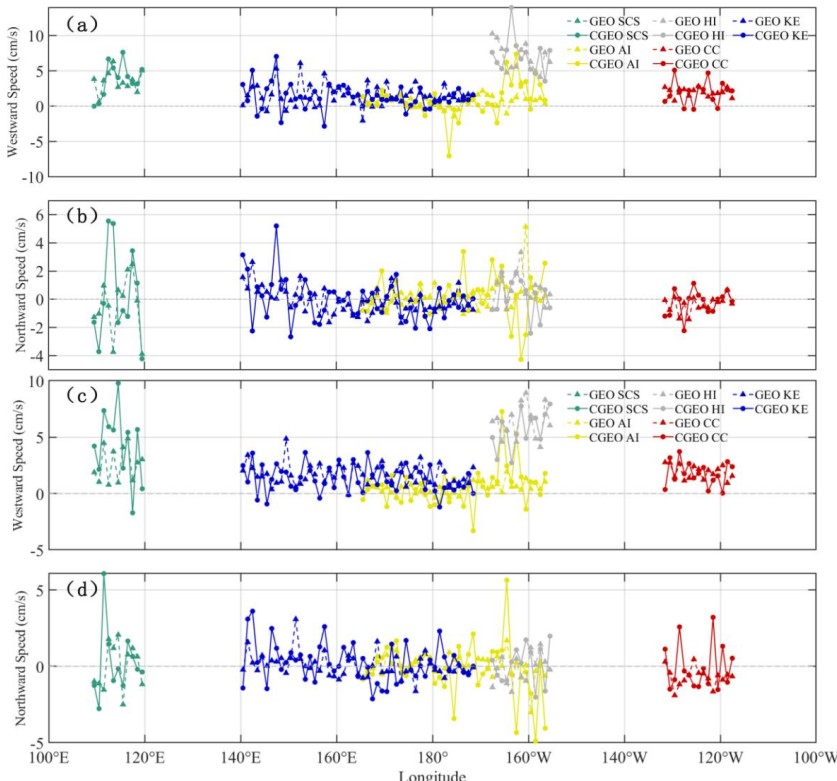

**Figure 14 Same as Figure 13, but showing the variations by longitude.**

**3.5 Vorticity and Strain Rate**

Vorticity describes the rotation of a fluid. Theoretically, the vorticity of an eddy reaches its maximum at the center and decreases gradually with increasing radial distance, diminishing to zero at the boundary. Thus, the sign of the normalized relative vorticity can be used to distinguish between cyclonic and anticyclonic eddies. As shown in Fig. 15a, in the SCS region, the normalized vorticity



under GEO balance increases rapidly during the generation phase, significantly exceeding that under

CGEO balance. In the CC region (Fig. 15b), the variation trends of GEO and CGEO are generally

similar. Throughout the generation phase, the vorticity increase in GEO is relatively small. However,

the normalized vorticity of GEO-AE rises more markedly than that of CGEO-AE, whereas the

normalized vorticity of CGEO-CE begins to decay directly. Notably, CGEO-CE shows a slight

increase during the dissipation phase. As can be seen from Fig. 15c, the overall vorticity curves display

more pronounced fluctuations. During the generation phase, the CE vorticity in both GEO and CGEO

increases slightly at comparable rates. The GEO-AE vorticity shows a mild increase, while CGEO-AE

experiences a slight decrease; nevertheless, the initial vorticity of GEO-AE remains higher than that of

CGEO-AE. By the maturation phase, mid-term CGEO-AE undergoes noticeable growth, which

continues into the dissipation phase. Eventually, GEO-AE dissipates to a lower value than CGEO-AE.

In the KE region (Fig. 15d), the CGEO curve exhibits a slowly decaying trend. However, CGEO

vorticity remains higher than GEO during both the initial generation and final dissipation phases. In the

HI region (Fig. 15e), the vorticity increase of CGEO during the generation phase is smaller than that of

GEO. In the dissipation phase, the decay rate of CGEO is also lower. Although CGEO vorticity

exceeds GEO at both the initial and final stages, its peak value during the maturation phase is lower

than that of GEO.

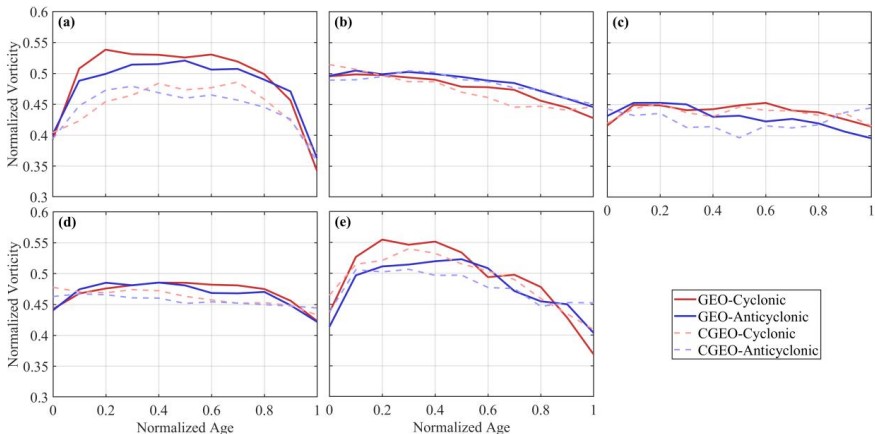

**Figure 15 Temporal evolution of the mean normalized relative vorticity for the (a) SCS, (b) CC, (c) AI, (d)
KE, and (e) HI regions.**



The eddy strain rate quantifies the capacity of eddy to distort and mix the surrounding ambient fluid. This parameter is key to understanding how eddies influence material transport, energy transfer, and the triggering of various secondary processes in the ocean. In the SCS region (Fig. 16a), CGEO

exhibits a higher overall strain rate than GEO, but a much lower rate of increase during the dissipative stage. In the CC region (Fig. 16b), both GEO and CGEO strain rate curves follow a similar declining trend, though CGEO shows a smaller increase during dissipation. In the AI region (Fig. 16c), the AE curve is more curved than the CE curve. Notably, the CGEO curve falls faster but recovers more slowly than the GEO curve. In the KE region (Fig. 16d), the GEO curve forms a shallow concave shape,

while the CGEO curve is stable initially then rises during dissipation. By comparison, CGEO declines more sharply and recovers more slowly. With reference to Fig.16e , the curve shows an approximately concave decline in the HI region. The CGEO curve exhibits smaller fluctuations, while the GEO curve shows more curvature. The rates of decrease and increase during the generation and dissipation stages are higher in GEO than in CGEO. The GEO-AE curve experiences an upward trend followed by a

downward fluctuation during the eddy maturation phase.

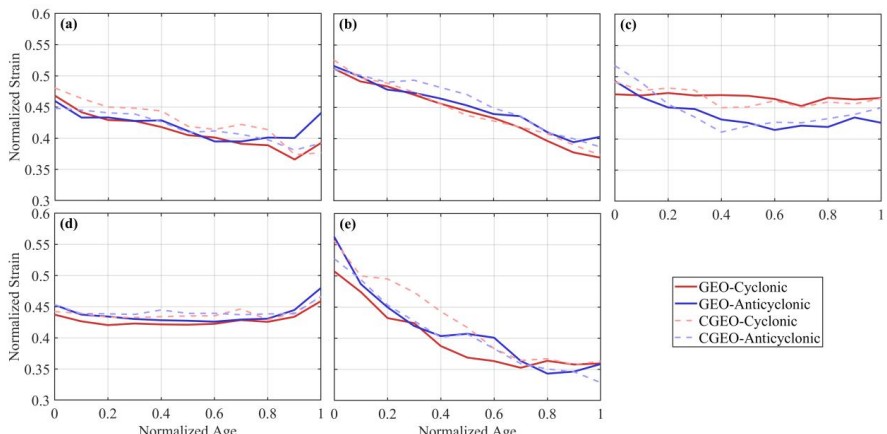

**Figure 16 Temporal evolution of the mean normalized strain rate for the (a) SCS, (b) CC, (c) AI, (d) KE, and (e) HI regions.**



## 4 Discussion


Based on satellite altimetry data and an automated eddy detection algorithm, a mesoscale eddy database for the North Pacific has been compiled. This enables a systematic comparison of statistical characteristics, including spatial distribution, temporal evolution, morphological features, and kinematic properties, between GEO and CGEO eddies. The comprehensive analysis reveals significant

differences in these aspects under the two dynamical frameworks. Case studies of individual eddies provide deeper insights into their lifecycle evolution (Fang and Morrow, 2003). To investigate the influence of curvature on the lifespan and physical mechanisms of individual eddies, one CE and one AE were selected from each of the five study regions in the North Pacific. A comparative analysis was performed focusing on the percentage difference between CGEO and GEO flow velocities, the

percentage difference in eddy enstrophy, the percentage difference in strain rate, and the evolution of eddy kinetic energy.

### 4.1 Aleutian Islands

The AI region, being the highest-latitude area among the five study regions, exhibits the largest Coriolis parameter, approximately $1.2 \times 10^{-4}$ rad/s. The selected CE in this region is centered between

53–53.5 °N, with Rossby numbers of 0.030 under GEO balance and 0.027 under CGEO balance. The minimum velocity difference percentage within the CE (Fig. 17a) is -5.74%, while the maximum percentage differences in eddy enstrophy (Fig. 17b) and strain rate (Fig. 17c) reach -4.27% and -53.66%, respectively.

The selected AE, located between 54.5–55 °N, shows Rossby numbers of 0.027 under GEO balance

and 0.028 under CGEO balance. Its maximum velocity difference percentage (Fig. 17a) is 5.57%, with maximum percentage differences in eddy enstrophy (Fig. 17b) and strain rate (Fig. 17c) of 8.42% and 46.68%, respectively. Curvature effects are less pronounced in the AI region, where centrifugal contributions are minor compared to the stronger Coriolis force.

Figures 17d and 17e illustrate the evolution of CGEO and GEO EKE for AE and CE, respectively. For

the anticyclonic eddy, both the growth and decay rates of CGEO-EKE exceed those of GEO-EKE, and the overall CGEO-EKE values are consistently higher, suggesting that AE influenced by curvature dissipate more readily. In contrast, the CE shows a smoother EKE evolution under the CGEO framework, characterized by smaller fluctuations and greater stability.





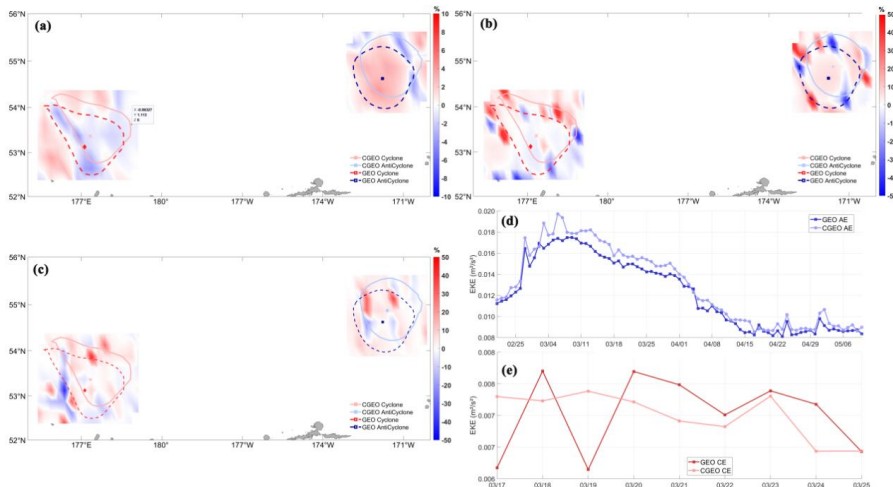


**Figure 17 Snapshot of cyclonic and anticyclonic eddy cases in the AI region (March 25, 2018): (a) Percentage difference between CGEO and GEO velocity, (b) Percentage difference between CGEO and GEO enstrophy, (c) Percentage difference between CGEO and GEO strain rate; (d) Evolution of EKE for a selected AE under CGEO and GEO balance (February 22 to May 10, 2018, where the light blue solid line**

**represents GEO-EKE, and the blue solid line represents CGEO-EKE, in units of m²/s²), (e) Evolution of EKE for a selected CE under CGEO and GEO balance (March 17 to 25, 2018, where the light red solid line represents GEO-EKE, and the red solid line represents CGEO-EKE, in units of m²/s²).**

### 4.2 California Coastal Current region

The California Coastal Current(CC) is a southward-flowing eastern boundary current of the North

Pacific Ocean, characterized by cool temperatures and a highly meandering flow. The selected CE in the CC region is centered between 27 °N and 28 °N, where the Coriolis parameter is approximately $6.7 \times 10^{-5}$ rad/s. Under GEO balance, its Rossby number is 0.032. After applying cyclogeostrophic correction, the value decreases to 0.027. As shown in Figure 18a, the minimum velocity difference percentage reaches –6.19%; Figure 18b shows that the maximum eddy enstrophy difference is –7.58%;

and Figure 18c indicates that the maximum strain rate difference reaches –23.58%. With the inclusion of curvature effects, the movement of the CE slows down, and the system exhibits an overall trend toward greater stability.

The AE selected in this region is centered between 25-25.5 °N, with a Coriolis parameter of approximately $6.2 \times 10^{-5}$ rad/s. Under GEO balance, the GEO Rossby number is 0.067; after

cyclogeostrophic correction, it decreases slightly to 0.066. Within the AE, the maximum velocity



difference percentage reaches 7.58% (Fig. 18a), while the maximum eddy enstrophy difference is 10.33% (Fig. 18b). As shown in Fig. 18c, the maximum difference in CGEO strain rate reaches 31.63%. The absolute values of all these difference percentages are larger for the AE than for the CE. This indicates that, following cyclogeostrophic correction, the AE exhibits greater instability, higher

susceptibility to deformation, and enhanced dynamic intensity compared to the CE. Therefore, the cyclogeostrophic correction has a more substantial modulating influence on the dynamical characteristics of AEs.

A comparison of the temporal evolution of the EKE under CGEO and GEO frameworks reveals systematic biases in the traditional GEO balance, showing that the EKE of the CE is overestimated (Fig.

18d), whereas that of the AE is underestimated (Fig. 18e). Further analysis shows that the decay rate of CGEO EKE for the AE is significantly faster than its geostrophic counterpart (Fig. 18e). In contrast, the decay amplitudes of CGEO and GEO EKE for the CE remain relatively similar (Fig. 18d).



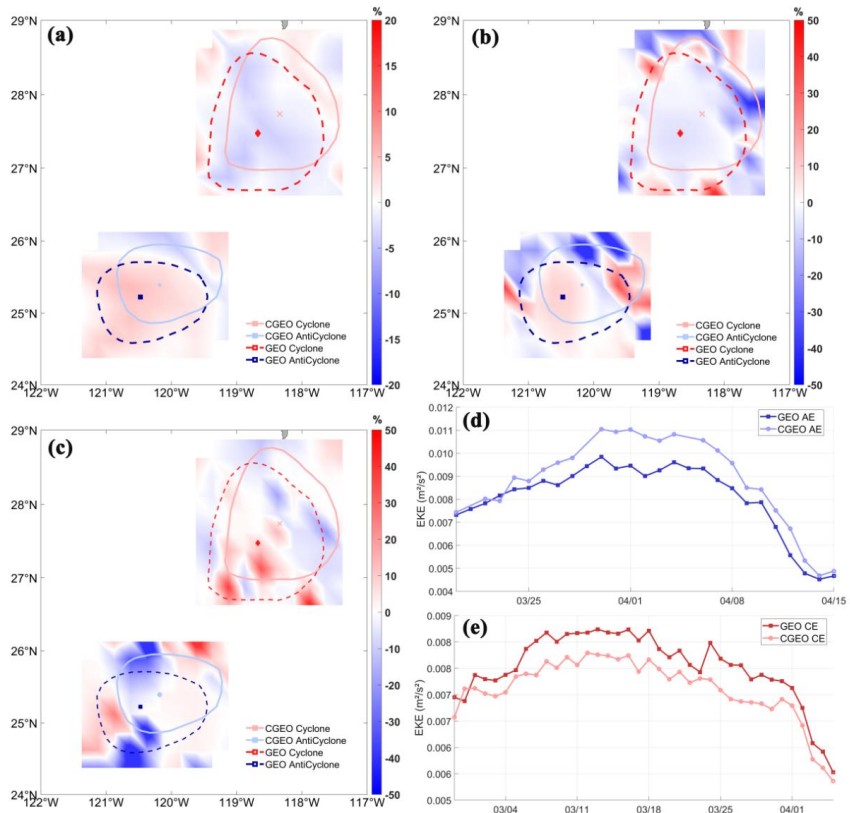

**Figure 18 Snapshot of cyclonic and anticyclonic eddy cases in the CC region (March 25, 2018): (a) Percentage difference between CGEO and GEO velocity, (b) Percentage difference between CGEO and GEO enstrophy, (c) Percentage difference between CGEO and GEO strain rate; (d) Evolution of eddy kinetic energy for a selected anticyclonic eddy under CGEO and GEO balance (March 20 to April 15, 2018), (e) Evolution of eddy kinetic energy for a selected cyclonic eddy under cyclogeostrophic and geostrophic balance (February 27 to April 15, 2018).**

### 4.3 Kuroshio Extension

The Kuroshio Extension (KE) is the eastward-flowing continuation of the Kuroshio Current, which serves as the western boundary current of the North Pacific Subtropical Gyre. It is a highly dynamic and unstable jet, characterized by large meanders, energetic eddies, and strong decadal variability. The selected CE in this region is located between 33-34 °N, with a Coriolis parameter of approximately $8 \times 10^{-5}$ rad/s. Under GEO balance, the Rossby number of the CE is 0.133, and under CGEO balance, it is 0.105. The maximum velocity difference percentage (Fig. 19a) reaches -20.67%, the maximum eddy enstrophy difference percentage can reach -31.36% (Fig. 19b), and the strain rate difference percentage





is less than -50% (Fig. 19c). In the KE, a region known for strong current-eddy interactions, the

inclusion of curvature effects results in a slower flow velocity within the CE compared to the GEO

balance. This reduced velocity weakens both eddy-eddy and eddy-current interactions, thereby

enhancing the stability of the CE.

The selected AE is centered between 36-37 °N with a Coriolis parameter of approximately $8.7 \times 10^{-5}$

rad/s. Under GEO balance, the Rossby number is 0.120, which increases slightly to 0.121 under CGEO

balance. In the AE, the maximum velocity difference reaches 23.57% (Fig. 19a), the maximum eddy

enstrophy difference is 20.42% (Fig. 19b), and the strain rate difference exceeds 50% (Fig. 19c). In the

Kuroshio Extension, a region characterized by a strong background flow and pronounced shear, the

CGEO strain rate of the AE is significantly greater than its GEO counterpart. Following

cyclogeostrophic correction, the AE exhibits a higher tendency for deformation and more active energy

exchange. These results indicate that cyclogeostrophic dynamical correction exerts a more substantial

influence on the dynamical properties of AEs..

A comparison of the EKE evolution of the CE (Fig. 19d) and AE (Fig. 19e) under CGEO and GEO

balance conditions shows that GEO balance overestimates the EKE of the CE while underestimating

that of the AE. Compared to GEO balance, the CGEO-EKE of the AE decays more rapidly, whereas

the decay amplitudes of the CE are similar under the two balance conditions. This confirms that

curvature has a greater impact on the stability of the AE.



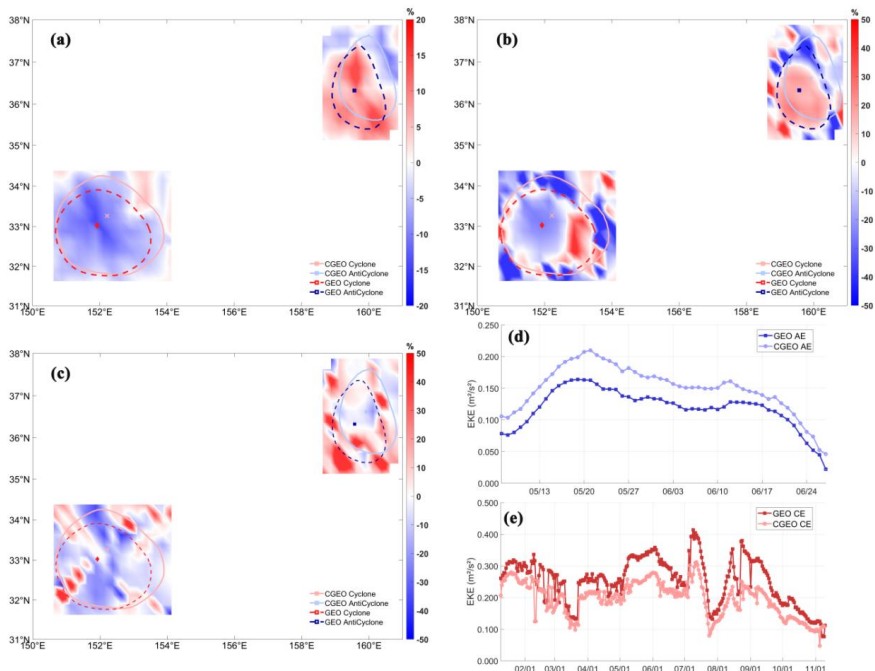

**Figure 19** Snapshot of CE and AE cases in the KE region (May 25, 2018): (a) Percentage difference between CGEO and GEO velocity, (b) Percentage difference between CGEO and GEO enstrophy, (c) Percentage difference between CGEO and GEO strain rate; (d) Evolution of EKE for a selected AE under CGEO and GEO balance (May 7 to June 15, 2018), (e) Evolution of EKE for a selected CE under CGEO and GEO balance ((January. 1-DecNovenmber. 10, 2018).

## 4.4 Hawaiian Islands

The HI region is a low-latitude oceanic area, with the selected eddies located between 18–19 °N, where the Coriolis parameter is approximately $4.6 \times 10^{-5}$ rad/s. For the CE, the Rossby numbers under GEO and CGEO balance are 0.118 and 0.114, respectively. Within the CE, the maximum velocity difference percentage (Fig. 20a) reaches 8.64%, the maximum eddy enstrophy difference percentage (Fig. 20b) can reach -25.74%, and the maximum strain rate difference percentage (Fig. 20c) is more than -50%. For the AE the Rossby numbers under GEO and CGEO balance are 0.123 and 0.106, respectively. Within the AE, the maximum velocity difference percentage (Fig. 20a) reaches 26.42%, the maximum eddy enstrophy difference percentage (Fig. 20b) is 43.37%, and the maximum strain rate difference percentage (Fig. 20c) is greater than 50%.

This sensitivity of the cyclogeostrophic correction to the energy state of AEs highlights its fundamental importance in regulating their dynamical properties. A comparison of the CGEO and GEO evolution



processes between the AE (Fig. 20d) and the CE (Fig. 20e) reveals distinct behavioral contrasts. In the

AE, both the growth and decay rates of CGEO-EKE are higher than those under the GEO framework,

and its peak EKE value also surpasses the GEO estimate. Conversely, the CE exhibits the opposite

pattern, in which GEO-EKE shows higher growth and decay rates along with a larger peak value than

its CGEO counterpart. These findings indicate that curvature effects promote dissipation in AEs while

simultaneously stabilizing CEs.

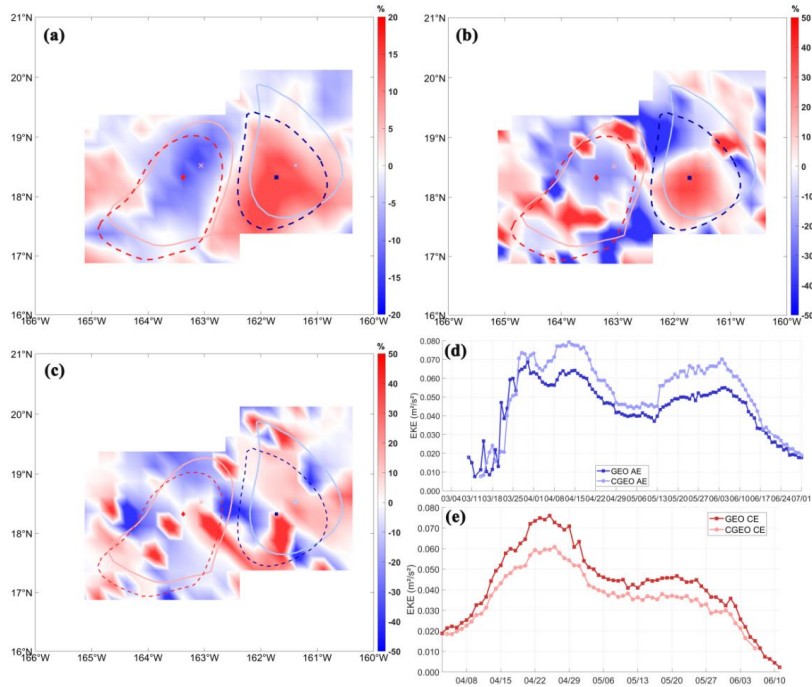

**Figure 20 Snapshot of CE and AE cases in the HI region (May 25, 2018): (a) Percentage difference between**
**CGEO and GEO velocity, (b) Percentage difference between CGEO and GEO enstrophy, (c) Percentage**
**difference between CGEO and GEO strain rate; (d) Evolution of EKE for a selected AE under CGEO and**
**GEO balance (May 10 to July 1, 2018), (e) Evolution of EKE for a selected CE under CGEO and GEO**
**balance (April 4 to June 1, 2018).**

**4.5 South China Sea**

The South China Sea , located at the lowest latitude among the five study regions, has a relatively small

Coriolis parameter of approximately $2.4 \times 10^{-5}$    rad/s. The selected CE is centered between 10°N and

11°N. Under GEO balance, its Rossby number is 0.110, which slightly decreases to 0.108 after

cyclogeostrophic correction. As shown in the figures, the maximum velocity difference percentage



reaches –24.38% (Fig. 21a), the maximum eddy enstrophy difference is –24.5% (Fig. 21b), and the

maximum strain rate difference is more than –50% (Fig. 21c). These results suggest that when

curvature effects are taken into account, the CE moves more slowly relative to the GEO balance, and

both eddy-eddy and eddy-flow interactions are weakened, thereby enhancing the stability of the

CE..The selected AE is centered between 9°N and 10°N. Under GEO balance, its Rossby number is

0.193, which decreases to 0.171 after cyclogeostrophic correction. Within the AE, the maximum

velocity difference percentage reaches 36.91% (Fig. 21a), and the maximum eddy enstrophy difference

is 53.86% (Fig. 21b). As shown in Fig. 21c, the CGEO strain rate exceeds the GEO value by more than

50%, indicating that the AE becomes more prone to deformation and exhibits enhanced dynamic

intensity following cyclogeostrophic correction. Furthermore, Fig. 21e reveals that the decay rate of

CGEO-EKE for the AE is substantially higher than that of its GEO counterpart, whereas the decay

amplitudes of the CE under the two frameworks remain relatively similar (Fig. 21d).


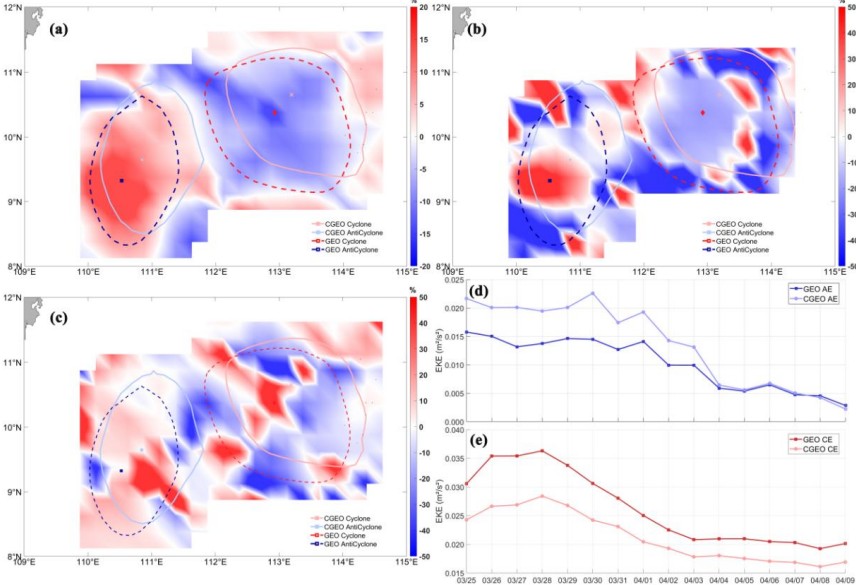

**Figure 21    Snapshot of CE and AE cases in the SCS (March 25, 2018): (a) Percentage difference between CGEO and GEO velocity; (b) Percentage difference between CGEO and GEO enstrophy; (c) Percentage difference between CGEO and GEO strain rate; (d) Evolution of EKE for a selected AE under CGEO and GEO balance (March 25 to April 9, 2018); (e) Evolution of EKE for a selected CE under CGEO and GEO balance (March 25 to April 9, 2018).**



**5 Conclusion**

Eddies are a key component of oceanic currents and a central focus in the study of ocean dynamic processes. The classical GEO balance theory neglects the centrifugal force of seawater motion, leading

to biases in sea surface velocities derived from altimeter-based sea surface height fields in curved flow regions. Based on GEO from satellite altimetry and CGEO corrected via an iterative method, this study statistically analyzes the differences in eddy characteristics across five regions of the North Pacific. It focuses on the dynamical behaviors of CEs and AEs under the two balance conditions, and systematically investigates the regulatory effects of the curvature effect on eddy energy, structural

features, and evolutionary processes.

A comparative analysis of the spatial distribution of EKE difference percentages and interannual evolution characteristics under the two balance conditions reveals that the difference patterns for eddies with negative and positive relative vorticity are completely opposite: after cyclogeostrophic correction, the EKE of CEs weakens, while that of AEs strengthens. On the temporal scale, cyclogeostrophic

correction is more significant for the geostrophic flow field of AEs. Useing the SCS as an example, the evolution of normalized EKE with the normalized eddy lifetime parameter shows that CGEO EKE increases faster during the growth stage, decays more slowly during the decay stage, and has higher energy retention. Meanwhile, the variation range of CGEO EKE is significantly smaller than that of GEO, confirming the regulatory role of curvature correction in balancing eddy energy

distribution.From the perspective of lifecycle stages, the differences in energy evolution are distinct: during the generation and decay stages, the energy increase and decrease rates of CGEO eddies are generally lower than those of GEO estimates; in the mature stage, the two tend to converge. For AEs, cyclogeostrophic correction significantly accelerates energy decay, and the curvature effect intensifies their instability, thereby promoting the dissipation process.

A comparison of eddy statistical characteristics shows that GEO identify 35.65% more eddies than CGEO. However, their spatial distributions of eddy numbers are strongly consistent: eddy activity is weaker and eddy counts are lower in coastal areas and near the equator. CGEO eddies have a larger average radius and shorter lifetimes. CE lifetimes are generally longer than AE lifetimes, indicating that curvature effects enhance eddy deformability and energy dissipation tendencies, with a more

significant impact on AEs.The spatial distribution patterns of eddy radius are highly similar between



the two datasets. The evolution of radius normalized by eddy lifetime follows the same trend as EKE evolution, showing an upward convex curve. During the generation phase, CGEO eddy radius increases more slowly than GEO eddy radius. During the decay phase, GEO eddy radius decays faster than CGEO eddy radius.In addition, GEO identify more eddies than CGEO during both generation and dissipation phases, but their interannual variations and spatial distribution characteristics are similar. In most regions, eddy westward propagation speed decreases with increasing latitude. The speed variation curve of CGEO eddies fluctuates more significantly than that of GEO eddies.Curvature correction generally has a stronger effect on CEs than on AEs. Differences in eddy enstrophy and strain rate further confirm that anticyclonic eddies are more sensitive to curvature, with enhanced dynamical characteristics after correction.

An analysis of regional eddy case differences reveals the latitudinal dependence of curvature effects: cyclogeostrophic correction has a particularly significant impact in low-latitude regions (such as the SCS and HI) and strong current regions (such as the KE). This indicates that regions with small Coriolis parameters and high Rossby numbers are affected more strongly by curvature effects. In contrast, the AI region— the highest-latitude study area—has the largest Coriolis parameter, leading to relatively weak centrifugal force contributions, smaller curvature effects, and smaller correction magnitudes.From the perspective of eddy types, cyclogeostrophic correction plays a more critical role in shaping the dynamical properties of AEs. Curvature effects increase AE EKE but reduce their stability, making AEs more prone to deformation and dissipation. For CEs, however, curvature effects slow movement, weaken eddy-eddy and eddy-current interactions, and significantly enhance stability.

**Code, data, or code and data availability**

The AVISO/DUACS 2018 altimeter data used in this study are available from the Copernicus Marine Service (https://cds.climate.copernicus.eu/datasets/satellite-sea-level-global?tab=download). The product can access from https://doi.org/10.24381/cds.4c328c78. Downloading altimeter satellite gridded data for free need to follow the data policy of the Copernicus Marine Service. The cyclogeostrophic current data employed in this study were derived from the geostrophic currents calibrated by corresponding author.



**Author contributions**

Conceptualization, Y.C. and X.Z.; methodology, Y.C.,L.L. and X.Z;formal analysis, Y.C. and X.Z; data curation, X.Z.; writing―original draft preparation, X.Z.,Y.C.,L.L.,Y.D. and R.L.; writing―review and editing, X.Z.,Y.C.,L.L.,Y.D.,R.L. and Z.Y.; .; funding acquisition, X.Z.,Y.C.,L.L.,Y.D.,R.L. and Z.Y.. All authors have read and agreed to the published version of the manuscript.

**Competing interests**

The contact author has declared that none of the authors has any competing interests.

**Disclaimer**

Publisher's note: Copernicus Publications remains neutral with regard to jurisdictional claims made in the text, published maps, institutional affiliations, or any other geographical representation in this paper. While Copernicus Publications makes every effort to include appropriate place names, the final responsibility lies with the authors. Views expressed in the text are those of the authors and do not

necessarily reflect the views of the publisher.

**Financial support**

This research was funded by the National Natural Science Foundation of China, grant number 42306028; the Natural Science Foundation of the Jiangsu Higher Education Institutions of China, grant number 23KJB170005;the Research Startup Foundation of Jiangsu Ocean University, grant number

KQ22012

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
