# Peer review of "Comparative Mesoscale Eddy Dynamics under Geostrophic versus Cyclogeostrophic Balance from Satellite Altimetry"

_EGUsphere, 2025_

## Referee Comment (RC1)

**Review on the paper**

Ĉomparative Mesoscale Eddy Dynamics under Geostrophic versus Cyclogeostrophic Balance from Satellite Altimetry

by Zhu, X., Cao, Y., Liu, L., Deng, Y., Liu. R. and You, Z.

The paper presents a comparisons of an eddy census and subsequent analysis using the same eddy identification tool but using either a velocity obtained by geostrophic balance or cyclogeostrophic balance from SSA. The latter being slightly more accurate.

The paper has merits and is of interest to physical oceanographers. It would nonetheless benefit from some revisions.

- Eddy census/analysis strongly depend on the criteria used to define/detect eddies: the method used and field used as well as the value of any threshold used.
  - The identification tool iused is not sufficiently descrived in the text, leaving 'the fourth constraint' a mystery.
  - Are the differences observed using the two different balances larger than differences that would be obtained using the same balance but different eddy detection tool?
  - The inclusion of the centripetal acceleration (which has the same sign regardless of the orientation of the eddy rotation) induces an asymmetry between cyclones and anticyclones which is not present in the geostrophic balance. Can the authors explain some of the differences observed by this?
- Section 3 organises the discussion by physical quantities and analyses there each region while the discussion (section 4) organises the discussion by region. This change is organisation somehow breraks the flow of reading. Is it necessary?
- Parts of the text can be improved. Please see below some specific points

1. Throughout the paper 'curvature effect' is ambiguous and too vague. I suggest the authors use 'centrifugal force' or 'centripetal acceleration'. Curvature effects could refer to the Earth's curvature. i.e. the latitudinal variation of the Coriolis parameter.

2. l 27: '... vorticity is more stable' vorticity is a quantity. It cannot be stable or unstable. An eddy can.

3. l 30: The statement 'translate faster' is more than surprising. I understand eddies can have slightly different characteristics if analysed using two different balances but the statement suggests they are not even at the same location... Surely the two analysis analyse the same SSA and the latter itself would give the location of eddies.

4. l 44: The authors mention the 3D structure of eddies, but only analyse SSA so do not have access to the 3D strucuture (unless they make further assumptions).

5. l46-49: It would be mnuch simpler (and clearer and shorter) to state that cyclones rotate in the same direction as the Earth and anticlyone in the opposite direction.

6. l 78: what is meant by 'significant periods'

7. l 89-90: the statement 'These biases... wind stress' must be further explained.

8. l 91-92: is somehow a trivial statement as cyclongeosptrophic balance is more accurate. Is it needed (at least in this form)?

9. l 95-96: what is meant by 'theoretical'?

10. l 120: velocity anoalies: these have not been defined. Why 'anomalies'? Also $EKE$ is not an energy (only its voluem integral - including multiplying by density - is). It is an EKE density (assuming it is measured on a isopycnal and the density is ignored for simpicity)

11. Equation (2) does not define a percentage (unless it is multiplied by 100).

12. l 123-125: The authors can simplify by simply stating that the index $i$ refers to quantities obtained through cyclogeostrophic balance and $g$ by geostrophic balance.

13. l 127-128 Explain the statement 'Entrophy...' and how it 'account for dissipative effets'. This seems incorrec

14. l 169: I assume the authors mean geostrophic velocities not satellite altimetry data (same l84 etc...).

15. 175-176 'due to wind stress and current shear': what is the link with cyclogeostrophic balance?

16. l 330-333: There are many mechanisms which can lead to the destruction of an eddy: instability, interaction with nathymetry, coast, other eddies and currents. Why is (turbulent) dissipation the only mechanism singled out?

17. Fig 8 and text: explain how the age is normalised

18. l 537: include 'local' before 'rotation' otherwise the statement is incorrect. Does the eddy description that follows agree with the criteria used to detect eddies (I don't think so). Moreover, it is questionable. For example. one would most likely to identify eddies using potential vorticity, not (the vertical component of) the (relative) vorticity.

**Minor points**

- l 11: 'among' → between
- l 12: 'induced owing to' → 'induced by'
- l 15: remove 'their' before 'cyclogeostrophic'
- l 19: 'cyclogeostrophic correction' → 'cyclogeostrophic balance' (as cyclogeostrophic includes both the 'correction (centripetal acceleration) and the leading order geostrophic balance)
- l 24: 'exhibiting opposite biases' is too vague to be informative
- l 44: 'evolutionary behavior' → 'evolution'
- l 58-59: 'a region of highly active mesoscale eddy activity' deos not read well (and acticity is unexplained/undefined - same l 70)
- l 84: remove 'assumption'
- l 86: 'classical' is unnecessary
- Eq (5) Ro is undefined. Is $f$ constant or measured locally?
- l 190: 'pronounced' → 'larger'
- l 191: remove 'percentage' (unnecessary)
- l 198: ' pronounced divergence' → 'larger difference' (divergence should be reserved for the mathematical oerator in an oceanogrphic paper)
- l 224: 'evolutionary pattern' → 'evolution'
- l 237: What is the 'mature phase'. 'close' → 'close to each other'
- l 238: 'decline' → decrease
- l 241: 'lower than' → less than

---

## Referee Comment (RC2)

A review of

*Comparative Mesoscale Eddy Dynamics under Geostrophic versus Cyclogeostrophic Balance from Satellite Altimetry*

by

Xinman Zhu, Yuhan Cao, Linxiao Liu, Yigang Deng, Ruixiang Liu, and Zhiwei You

—

In this study the authors have explored how mesoscale eddy properties differ when considering cyclogeostrophic balance compared to geostrophic balance. They argue that the long-standing geostrophic balance assumption does not fully describe mesoscale eddy currents and a recalibration towards cyclogeostrophic balance should be considered. The authors consider separate regions across the North Pacific, motivated to improve their understanding of mesoscale currents across the Kuroshio Extension and wider afield. To conduct their analysis, the authors make use of satellite altimetry data at a horizontal resolution of 0.25 degrees. This data product is then used to compute the geostrophic and cyclogeostrophic eddy field, of which they acquire from Cao et al., (2023). Thereafter, a series of figures and analyses are presented that give the reader a strong sense that this change in balance can be important.

As the reader progresses through the paper it is clear that there are a number of things that could be improved. Although the results do demonstrate changes to eddy properties such as EKE and eddy distributions, the authors provide limited discussion on the processes that cause these changes. I think the authors could do a better job to tie in wider significance to these findings e.g. through process studies and the implications for say Earth system modelling - tuning of models through surface EKE, for example.

Overall, I believe that a study of this scope does fit into the Ocean Science journal and would be of interest to the wider community. However, there are a number of things (see below) that need addressing before it can be considered for publication. I recommend major revisions before publication. My comments are in no particular order.

**Comments**

- The reader could be helped by the inclusion of the geostrophic and cyclogeostrophic balance equations, a discussion of the differences, and a schematic to illustrate this. This could then be used to explain some of the differences in the results e.g. why the geostrophic assumption is suitable for the ACE EKE in CC region (Fig 1) but not in the KE region. Are there larger pressure/centrifugal differences? And on line 196 'Thus the discrepancy…' could give further insight here by including equations and discussion.

- The 'Discussion' section appears to be a continuation of results, focusing on case studies of ACE and CE in each region. Should a separate section within the results be made called 'Case Studies'? I would not put new results in a discussion section.

- In the abstract there is mention of a few process changes but no actual discussion of this in the paper. Please consider what is written in the abstract should be discussed in the paper. For example, 'possess stronger potential energy'. I think this line refers to the 'Discussion' section and Figure 18 (?), but there is no discussion of this in the paper. I would disagree that the eddy possesses stronger potential energy. Is the eddy barotropic, is it baroclinic? The sea level anomaly does not change, and no subsurface examination is carried out, so it is hard to say this.

- There are a number of places throughout the paper that could benefit by further explanation. Some examples, 'eddy generation enhanced due to upwelling' (line 297), and 'AE influenced by curvature dissipate more readily' (line 607). Are there any process based studies that show these to be the case?

- It is interesting that the Kuroshio Extension has the largest eddy radii (Fig 5), could there be a reason for this?

- The inclusion of the case studies is to try and draw some physical understanding from the results? Maybe this needs to be clear.

- At the end of the Introduction section, you could outline your paper e.g. 'In section X we will discuss the results, and in section Y …'. This will solve my above point.

- Stegner and Dritschel (2000) [A Numerical Investigation of the Stability of Isolated Shallow Water Vortices] looked at the stability of ACE and CE in cyclogeostrophic balance; maybe this paper will help to add further support and locate more citations.

- I somewhat understand the inclusion of strain rate and enstrophy, but feel it should probably be omitted. In 2D turbulence, dissipation is proportional to enstrophy, and so you could get some understanding from this. The strain rate similarly suggests dissipation but in 3D turbulence, which the 0.25 degree altimeter is never going to capture. Have there been studies that have looked at turbulence using satellite altimeter data? They are probably heavily constrained by resolution. An alternative could be to include the wind stress curl and Ekman pumping, but you then need a wind stress data product… you could look at spatial changes in EKE?

- The results are often quite long winded, could they be broken up and focus given to the most significant finding?

- Could other eddy census studies be discussed further e.g. Chelton et al (2011), Chen et al (2019) 10.1029/2019JC014983

- How do you think your results would differ if you chose an alternative eddy detection algorithm? Would eddy radii change? Just something to think about.

**Specific comments**

- Line 22-23: eddy kinetic energy and Eddy Kinetic Energy - decide what to use.
- Line 24: 'Eddies predominantly …' this isn't new.
- Line 27: 'Vorticity is more stable' - the eddy is more stable by virtue of a smaller Rossby number?
- Line 31-32: Where do you look at eddy-eddy and eddy-current interactions?
- Line 50: space between 'kilometers(Chelton'.
- Line 76: 'where leeward… shows significant periods…' do you cycles/variability?
- Line 108: source for satellite data?
- Line 112: put 'data are accessible …' in data availability, and cite in text.
- Line 121: what is the velocity fluctuation? Spatial, temporal - 5,10 days?
- Line 123: 'percentage difference…' between geos and cylco?
- Eq 5 is the dynamical Rossby number.
- Line 166: 'andSCS'
- Line 180: 'This study compares…' say this earlier?
- Line 183: 'Satellite altimetry overestimates…' using geostrophic assumption?
- Line 190: 'Spatially, the difference…' could be reworded to clarify cyclo is more positive than geos over the domain?
- Figure 1: Label each figure panel with the region (save reader from scanning caption and back to panels), remove legends from all but panel b.
- Figure 2: Agulhas Insurgence?
- Figure 2: Why are there differences across regions?
- Figure 3: Regions are quite small, can these be made bigger?
- Line 360: Any studies showing why these eddies are larger? Can you cite any.
- Figure 9 and 10: Less eddies generated and less dissipated in cyclo. All regions are fairly similar, can the results be compressed into one figure, side by side with generation and dissipation? Feel free to change or not, just a suggestion. Could any other figures be condensed? Show total and then most significant finding?
- Line 468: Why are there less eddies generated and dissipated under cyclo balance?
- Line 486: Fig 11b and d should be Fig 12?
- Figure 12: More of a uniform dissipation of geos eddies in KE region but not in cyclo, why?
- Figure 13: Label left column CE and right column ACE? Seems the biggest fluctuations are in SCS, any idea why?
- Figure 15: Could be consistent with labelling? E.g. KE region is sometimes presented first, but here it is in panel d. Why are there such differences, even in geos and cyclo, this is where other literature could come in use.
- Figure 16: You could almost say the vorticity and strain are inversely related?
- Line 580: Change name of discussion? Do you need to include all five case studies here?
- Line 732: Could additional sources and references be included here to tie up your findings? What are the implications, bigger picture, takeaway for the reader?
- Line 780: Where do you discuss eddy-eddy interactions and how do you define this?

---

## Author Comment (AC1)

Response: Thank the reviewer for the time spent on our manuscripts. The comments and suggestions by the reviewer are of great helps for us to improve our manuscripts significantly. We have addressed all the comments one by one very carefully.

● Eddy census/analysis strongly depend on the criteria used to define/detect eddies: the method used and field used as well as the value of any threshold used.

Response: Thank you for your comments. We agree with your perspective. The fundamental issue in studying eddy activity lies in the proper definition of eddies and the achievement of automatic identification and tracking for mesoscale and submesoscale eddies. The detection method employed in this study is a vector geometry-based automatic eddy detection algorithm. Detailed description of this detection method can be found in the following reference, which we have now cited in the manuscript.(L142)

"Nencioli, F., Dong, C., Dickey, T., Washburn, L., & McWilliams, J. C. (2010). A vector geometry–based eddy detection algorithm and its application to a high-resolution numerical model product and high-frequency radar surface velocities in the Southern California Bight. Journal of atmospheric and oceanic technology, 27(3), 564-579."

– The identification tool iused is not sufficiently descrived in the text, leaving 'the fourth constraint' a mystery.

Response: Thank you for your comments. We have redescribed the detection method (L155-162).

The constraints require the specification of two parameters: parameter a applies to the first, second, and fourth constraints, while parameter b corresponds to the third constraint. Parameter a defines the spatial range, expressed in number of grid points, over which the increase in the magnitude of v along the east-west axis and of u along the north-south axis is examined. It also specifies the closed contour around the eddy center along which changes in the direction of velocity vectors are evaluated. Parameter b determines the size of the region, also measured in grid points, that is used to identify the local velocity minimum (Nencioli et al., 2010). To ensure global applicability, the parameters are set to a=4 and b=3 based on empirical optimization.

– Are the differences observed using the two different balances larger than differences that would be obtained using the same balance but different eddy detection tool?

Response:Thank you for your comments. Different detection algorithms exhibit variations in eddy identification. For instance, You et al. (2022) show that the total number of eddies in the GOMEAD dataset is 8% lower than that in the META dataset. This study focuses on exploring the differences between cyclogeostrophically corrected altimetry data and original altimetry data in the statistical analysis of eddies.

– The inclusion of the centripetal acceleration (which has the same sign regardless of the orientation of the eddy rotation) induces an asymmetry

between cyclones and anticyclones which is not present in the geostrophic balance. Can the authors explain some of the differences observed by this?

Response: Thank you for your thoughtful comments. For a more comprehensive theoretical discussion, please refer to the two earlier publications by the corresponding author (listed below). Here, we provide only a concise analysis on this point.

References:

1. Cao, Y., Dong, C., Qiu, Z., Bethel, B. J., Shi, H.; Lv, H., & Cheng, Y. Corrections of Mesoscale Eddies and Kuroshio Extension Surface Velocities Derived from Satellite Altimeters. Remote Sensing, 2023a, 15(1): 184.

2. Cao, Y., Dong, C., Stegner, A., Bethel, B. J., Li C., Dong J., Lü H., & Yang, J. Global Sea Surface Cyclogeostrophic Currents Derived from Satellite Altimetry Data. Journal of Geophysical Research: Oceans, 2023b, 128(1): e2022JC019357.

The classical geostrophic balance theory ignores the effect of centrifugal force on the real sea motion, which makes the flow field, especially true for curved flow in mesoscale ocean eddies, measured by the altimeter has certain errors under the assumption of geostrophic balance. The curvature effect of streamlines makes the geostrophic balance between the pressure gradient force and Coriolis force be corrected as the balance between pressure, Coriolis and centrifugal forces (Figure 1).

[Figure]

**Figure 1.** The force diagram of cyclonic eddy (CE) and anticyclonic eddy (ACE) under geostrophic and cyclogeostrophic conditions.

For surface currents, introducing geostrophic velocities, the cyclogeostrophic current equation in cartesian coordinates can be transformed as:

$$\begin{cases} u\dfrac{\partial u}{\partial x}+v\dfrac{\partial u}{\partial y}-fv=-g\dfrac{\partial \eta}{\partial x} \\ u\dfrac{\partial v}{\partial x}+v\dfrac{\partial v}{\partial y}+fu=-g\dfrac{\partial \eta}{\partial y} \end{cases} \Rightarrow \begin{cases} u\dfrac{\partial u}{\partial x}+v\dfrac{\partial u}{\partial y}-fv=-fv_g \\ u\dfrac{\partial v}{\partial x}+v\dfrac{\partial v}{\partial y}+fu=fu_g \end{cases} \tag{1}$$

where $u_g$ and $v_g$ are the geostrophic velocity anomalies of zonal and meridional currents, $u$ and $v$ are surface velocities of zonal and meridional currents, $\eta$ and $g$ are the sea surface height and gravitational acceleration parameter, respectively. In order to give the solution of the cyclogeostrophic velocity components to Eq.1 for mesoscale eddies and strong surface mean currents with different shapes, this study uses an iterative method. The method was applied to solve the momentum equation of ocean currents under cyclogeostrophic equilibrium conditions. This iterative scheme is given by

$$\vec{u}^{(n+1)} - \frac{\vec{k}}{f} \times \left( \vec{u}^{(n)} \cdot \nabla \vec{u}^{n} \right) = \vec{u}_{g} \tag{2}$$

where $\vec{k}$ is the vertical unit vector, $\vec{u}$ is the surface velocity vector, and $\vec{u}_{g}$ is the geostrophic velocity vector. The specific iterative method is as follows:

$$u^{n+1} = u_{g} - \frac{1}{f}(u^{n} \frac{\partial v^{n}}{\partial x} + v^{n} \frac{\partial v^{n}}{\partial y})$$

$$v^{n+1} = v_{g} + \frac{1}{f}(u^{n} \frac{\partial u^{n}}{\partial x} + v^{n} \frac{\partial u^{n}}{\partial y})$$

$$n = 0,$$

$$u = u_{g}, \quad v = v_{g}$$

$$n = 1, \tag{3}$$

$$u^{(2)} = u_{g} - \frac{1}{f}(u_{g} \frac{\partial v_{g}}{\partial x} + v_{g} \frac{\partial v_{g}}{\partial y})$$

$$v^{(2)} = v_{g} + \frac{1}{f}(u_{g} \frac{\partial u_{g}}{\partial x} + v_{g} \frac{\partial u_{g}}{\partial y})$$

......

● Section 3 organises the discussion by physical quantities and analyses there each region while the discussion (section 4) organises the discussion by region. This change is organisation somehow breraks the flow of reading. Is it necessary?

Response:Thank you for your comments. Section 3 lays a systematic and quantitative foundation by organizing results according to key physical quantities — such as EKE, radius, and amplitude — to first establish statistical differences between the GEO and CGEO datasets. This part addresses the "what," namely the overarching effects introduced by the cyclogeostrophic correction. As is well known, the generation and dissipation mechanisms of eddies are highly complex. Section 4 focuses

on the dynamical differences of individual eddies across five specific regions, examining their detailed evolution under varying conditions of background flow curvature and latitude (i.e., Coriolis parameter).

● **Specif comments**

1)Throughout the paper 'curvature effect' is ambiguous and too vague. I suggest the authors use 'centrifugal force' or 'centripetal acceleration'. Curvature effects could refer to the Earth's curvature. i.e. the latitudinal variation of the Coriolis parameter.

Response:Thank you for your comments. First, the relationship between streamline curvature and centrifugal force is explained here. From a dynamical perspective, when an object moves relative to another at a variable velocity, curvature arises due to the warping of spacetime. In accordance with the principle of equivalence in general relativity, the properties of spacetime in a gravitational field depend on the distribution of mass. The distribution of mass causes spacetime to become inhomogeneous, leading to its curvature. When an object possesses mass (and thus velocity), it bends spacetime; the more massive the object, the greater the curvature of spacetime. The streamline curvature theorem states that in a fluid flow with curved streamlines, the pressure on the convex (outer) side of the curve is higher than on the concave (inner) side.

This resulting pressure difference provides the centripetal force that acts on the fluid.

To clarify the terminology in the manuscript, we have replaced the ambiguous term "curvature effect" with the more precise expressions: "streamline curvature effect," "centrifugal force," and "cyclogeostrophic correction." (e.g., L603, 618, 622, 676,692...)

2)L27: '... vorticity is more stable' vorticity is a quantity. It cannot be stable or unstable. An eddy can.

Response:Thank you for your comment. We have corrected "vorticity" to "eddy".(L27)

3)L30: The statement 'translate faster' is more than surprising. I understand eddies can have slightly different characteristics if analysed using two different balances but the statement suggests they are not even at the same location... Surely the two analysis analyse the same SSA and the latter itself would give the location of eddies.

Response: Thank you for your valuable feedback. You are absolutely right that the phrasing "translate faster" is ambiguous. we have corrected the phrase "anticyclonic eddies translate faster" to "the flow velocity within anticyclonic eddies increases".(L30)

4) L44: The authors mention the 3D structure of eddies, but only analyse SSA so do not have access to the 3D strucuture (unless they make further assumptions).

Response: Thank you for your suggestion. You are correct that the mention of the 3D structure was only conceptual, as the analysis is indeed based on SSA data. We have revised this section accordingly.(L43)

5) L46-49: It would be mnuch simpler (and clearer and shorter) to state that cyclones rotate in the same direction as the Earth and anticlyone in the opposite direction.

Response: Thank you for your valuable suggestion. We agree on the importance of clear and precise expression. We have carefully revised the relevant section to ensure the language is both concise and unambiguous, thereby improving the overall clarity of the manuscript.(L46-48)

"In each hemisphere of the Earth, cyclones rotate in the same direction as the planet's rotation (counterclockwise in the North, clockwise in the South), while anticyclones rotate in the opposite direction. "

6) L78: what is meant by 'significant periods'

Response: Thank you for your valuable comment. We acknowledge that the phrase "significant periods" could be ambiguous, as it might be

misinterpreted in a strict statistical sense. Our intended meaning was to highlight that the periodic variations themselves are pronounced. We have revised this phrasing to accurately convey that the eddy kinetic energy exhibits strong or pronounced periodicities at approximately 60 and 100 days.(L74)

7) L89-90: the statement 'These biases... wind stress' must be further explained.

Response:Thank you for raising this important point. We agree that the statement regarding biases requires clarification. In the revised manuscript, we will expand the explanation to explicitly distinguish between the two types of biases mentioned. (L87-89)

These biases are distinct from the ageostrophic velocity components induced by surface wind stress, and it is mainly caused by the nonlinear term induced by the local curvature of the streamline.

8) L91-92: is somehow a trivial statement as cyclongeosptrophic balance is more accurate. Is it needed (at least in this form)?

Response: Thank you for this insightful comment. We have revised the sentence to eliminate the redundant phrasing, and now state more directly the essential role of the cyclogeostrophic balance in these regions. (L89)

9) L95-96: what is meant by 'theoretical'?

Response:Thank you for your question regarding the term "theoretical frameworks." We agree that this phrasing was ambiguous in the original manuscript.In the context of our introduction, "theoretical frameworks" specifically refers to the cyclogeostrophic balance theory. The intended meaning of the sentence is that while this theory has been successfully applied in several well-defined western boundary current regions (as cited), its practical implementation and validation in the more complex and variable flow environment of the North Pacific—the focus of our study—have not been extensively reported. This gap motivates our research. We have revised the text to clarify this point.  (L94-95)

10) L120: velocity anoalies: these have not been defined. Why 'anomalies'? Also EKE is not an energy (only its voluem integral - including multiplying by density - is). It is an EKE density (assuming it is measured on a isopycnal and the density is ignored for simpicity)

Response:Thank you for raising these important methodological points. Please find our detailed responses and the corresponding revisions below.

The velocity anomalies u' and v' correspond directly to the geostrophic velocity anomaly fields ugosa and vgosa from the AVISO data. These anomalies are calculated by subtracting the 1993-2012 climatological

mean geostrophic velocity from the daily observed velocity at each grid point.

We have revised the text to specify that the quantity defined by Eq. (1) is the Eddy Kinetic Energy per unit mass (EKE density), and we have added a note clarifying its relationship to total kinetic energy. This correction has been made in the Methods section. (L118, L121-123)

This represents the kinetic energy of transient eddy motions per unit mass. The total eddy kinetic energy of a volume would require integration of this quantity over that volume, including multiplication by the seawater density.

11) Equation (2) does not define a percentage (unless it is multiplied by 100).

Response:Thank you for your suggestion. We have corrected Equation (2) to properly represent the result as a percentage.

12) l 123-125: The authors can simplify by simply stating that the index i refers to quantities obtained through cyclogeostrophic balance and g by geostrophic balance.

Response:Thank you for your suggestion. We have noted your point. For consistency with the terminology established in our equations and the

broader literature we are aligning with, we prefer to retain the original phrasing in the text. (L125-128)

13) l 127-128 Explain the statement 'Entrophy...' and how it 'account for dissipative effets'. This seems incorrect.

Response: Thank you for your comment. Enstrophy is closely linked to the dissipation of turbulent kinetic energy and represents a fundamental variable in turbulence theory. Regions of high enstrophy typically correspond to intense eddy activities, shear instabilities, or turbulent dissipation processes. In oceanographic and atmospheric dynamics, it is commonly employed to analyze the generation, evolution, interactions, and dissipation of eddies. We have re-defined this parameter. (L129-131)

14) L169: I assume the authors mean geostrophic velocities not satellite altimetry data (same l84 etc...).

Response:Thank you for your comment. You are right to point out the need for clarity regarding the geostrophic balance context. We have revised both sections to explicitly state this and improve overall readability. (L182, L196)

15) L75-176 'due to wind stress and current shear': what is the link with cyclogeostrophic balance?

Response: Thank you for your comment.   wind stress curl and shear create background flow curvature, which amplifies geostrophic balance error and thus affects the cyclogeostrophic correction magnitude. We have clarified this in the text.(L189-190)

which enhance flow curvature and velocity gradients, the limitation of the geostrophic balance becomes more pronounced. Consequently,

16) L330-333: There are many mechanisms which can lead to the destruction of an eddy: instability, interaction with nathymetry, coast, other eddies and currents. Why is (turbulent) dissipation the only mechanism singled out?

Response:Thank you for this insightful and absolutely correct comment. We agree that singling out turbulent dissipation as the sole mechanism was an oversimplification. As you rightly point out, eddy destruction involves multiple processes.In the revised manuscript, we have modified the sentence (Lines 341-342) to reflect this complexity.

17) Fig 8 and text: explain how the age is normalised

Response:Thank you for your comment regarding the normalization of eddy age. The eddy age in Figure 8 (and related analysis) is normalized. Specifically, we normalize the surface eddies that have a lifespan longer than 7 days,the X-axis is divided by the lifespan of each individual eddy,

while the Y-axis variables are normalized by dividing each variable by its corresponding maximum value throughout the entire lifespan process, the entire lifespan of each individual eddy is normalized to 1. Each life stage is then defined based on intervals of this normalized age: generation (~0-0.1), intensification (~0.1-0.3), maturation (~0.3-0.8), and decay (>0.8). This method allows for the compositing and comparison of eddy properties across eddies with different absolute lifespans.We have added the requested explanation to the manuscript text (and/or figure caption) as suggested. (Lines 376-378,Lines 421-422)

We normalize the surface eddies that have a lifespan longer than 7 days. The X-axis is divided by the lifespan of each individual eddy, while the Y-axis variables are normalized by dividing each variable by its corresponding maximum value throughout the entire lifespan process. Each life stage is defined by normalized age: generation (~0-0.1), intensification (~0.1-0.3), maturation (~0.3-0.8), and decay (>0.8).

18) l 537: include 'local' before 'rotation' otherwise the statement is incorrect. Does the eddy description that follows agree with the criteria used to detect eddies (I don't think so). Moreover, it is questionable. For example. one would most likely to identify eddies using potential vorticity, not (the vertical component of) the (relative) vorticity.

Response:Thank you for your valuable comment. We have re-written this part. (L552-556)

For understanding vorticity, we referred to the following books.

Batchelor, G. K.: An introduction to fluid dynamics, Cambridge Univ. Press, https://doi.org/10.1017/CBO9780511800955, 2000.

**Minor points**

l 11: 'among' → between

Response: Thank you for your comment.We have corrected "among" to " between". (L12)

l 12: 'induced owing to' → 'induced by'

Response: Thank you for your comment.We have corrected "induced owing to" to " induced by". (L13)

l 15: remove 'their' before 'cyclogeostrophic'

Response: Thank you for your valuable comment. As suggested, we have removed the word "their" before "cyclogeostrophic" in the manuscript. (L15)

l 19: 'cyclogeostrophic correction' → 'cyclogeostrophic balance' (as cyclogeostrophic includes both the 'correction (centripetal acceleration) and the leading order geostrophic balance)

Response : Thank you for your comment.We have corrected "cyclogeostrophic correction" to " cyclogeostrophic balance". (L19)

l 24: 'exhibiting opposite biases' is too vague to be informative

Response:Thank you for your valuable comment. We have revised that section as suggested, which involved deleting the original sentence and rewriting it with a more detailed explanation to better support our methodology. (L21-24)

The eddy kinetic energy exhibits significant regional variability, with cyclonic and anticyclonic eddies showing opposite signs of bias. After applying the cyclogeostrophic correction, the EKE of cyclonic eddies weakens, whereas that of anticyclonic eddies strengthens.

l 44: 'evolutionary behavior' → 'evolution'

Response:Thank you for your comment.We have corrected "evolutionary behavior" to " evolution". (L43)

l 58-59: 'a region of highly active mesoscale eddy activity' deos not read well (and acticity is unexplained/undefined - same l 70)

Response:Thank you for your comment. We have removed the phrase "a region of highly active mesoscale eddy activity" and revised both sections accordingly. (L56 and L67-68)

a region of highly active mesoscale eddies.

l 84: remove 'assumption'

Response:Thank you for your valuable comment. As suggested, we have removed the word "assumption" in the manuscript. (L81-82)

l 86: 'classical' is unnecessary

Response:Thank you for your comment. As suggested, we have removed the word "classical" in the manuscript. (L84)

Eq (5) Ro is undefined. Is f constant or measured locally?

Response: Thank you for your comment. We have added the necessary definitions directly after Eq. (5): Ro is rossby number,and f is defined as the local Coriolis parameter, calculated at the relevant latitude. (L137-138)

==Ro is rossby number,and f is coriolis parameter which is measured locally==.

l 190: 'pronounced' → 'larger'

Response: Thank you for your comment.We have corrected "pronounced" to " larger". (L202)

l 191: remove 'percentage' (unnecessary)

Response: Thank you for your comment. As suggested, we have removed the word "percentage" in the manuscript. (L202)

l 198: ' pronounced divergence' → 'larger difference' (divergence should be reserved for the mathematical oerator in an oceanogrphic paper)

Response: Thank you for your comment.We have corrected " pronounced divergence" to " larger difference". (L210)

l 224: 'evolutionary pattern' → 'evolution'

Response: Thank you for your comment. We have corrected " evolutionary pattern " to " evolution". (L234)

l 237: What is the 'mature phase'. 'close' → 'close to each other'

Response:Thank you for your comment.Regarding "mature phase": We have revised it to "maturation" in line with the terminology used in the caption of Figure 2.The entire lifespan of each individual eddy is normalized to 1. Each life stage is defined by normalized age: maturation (~0.3-0.8). Regarding "close": We have changed it to the clearer phrase "close to each other."These changes improve the precision and readability of the text. (L248)

l 238: 'decline' → decrease

Response:Thank you for your comment.We have corrected "decline" to " decrease". (L249)

l 241: 'lower than' → less than

Response:Thank you for your comment.We have corrected "lower than" to "  less than". (L252)

---

## Author Comment (AC2)

Response: We sincerely thank the reviewer for the time and effort dedicated to evaluating our manuscript. The comments and suggestions provided have been immensely helpful in improving the quality of our work. We have carefully addressed each point raised in the comments.

**Comments**

● The reader could be helped by the inclusion of the geostrophic and cyclogeostrophic balance equations, a discussion of the differences, and a schematic to illustrate this. This could then be used to explain some of the differences in the results e.g. why the geostrophic assumption is suitable for the ACE EKE in CC region (Fig 1) but not in the KE region. Are there largerpressure/centrifugal differences? And on line 196 'Thus the discrepancy…' could give further insight here by including equations and discussion.

**Response 1**: Thank you for your comments. Following your suggestion, firstly, we have added the relevant formulas for the cyclogeostrophic balance in Section 2. Secondly, to describe and explain Figure 1 more clearly, we have reorganized the corresponding paragraph. (L202-206, L210-212, L227-229)

**2.2 Cyclogeostrophic balance equations**

In the context of an horizontal, stationary, and inviscid flow, the momentum equation which links SSH and surface horizontal currents $U$ is:

$$U \cdot \nabla U + fk \times U = -g \nabla \eta \tag{1}$$

where $f$ is the Coriolis parameter, $\vec{k}$ is a vertical unit vector, $\eta$ and $g$ are the sea surface height and gravitational acceleration parameter, respectively. For Mesoscale curved ocean currents, the nonlinear term induced by the local curvature of the streamline cannot be ignored. Introducing the geostrophic velocities $U_g$ from (1), this equation can be transformed to the form

$$\vec{U} - \frac{\vec{k}}{f} \times (\vec{U} \cdot \nabla \vec{U}) = \vec{U}_g \tag{2}$$

In cylindrical coordinates, this equation simplifies for the azimuthal velocities to the gradient wind equation (Knox and Ohmann, 2006):

$$\frac{V^2}{fR} + V = V_g \tag{3}$$

where $V$ is the azimuthal velocity, $V_g$ is the geostrophic velocity derived from the pressure gradient, and $R$ is the radius of curvature. This equation admits an analytical solution for $V$.

According to Eq.3, the cyclogeostrophic Rossby number can be given by:

$$Ro_c = \frac{V}{fR} = \frac{V_g - V}{V} \tag{4}$$

Analysis shows that the discrepancies between geostrophic and cyclogeostrophic currents are governed by curvature:

$$\kappa = \frac{f(V_g - V)}{V^2} \tag{5}$$

where, $\kappa$ is the curvature.

Additionally, the weaker Coriolis force in this region leads, according to Eq. 5, to a more significant cyclogeostrophic correction effect. Driven by the combined effects of the Coriolis force and prevailing wind belts, the North Pacific Subtropical Gyre exhibits basin-wide cyclonic circulation. Within this gyre, the CC region shows a notable prevalence of cyclonic eddies among long-lived surface eddy populations (Kurian et al., 2011). (L202-206)

This is primarily because the larger Coriolis force in high-latitude regions results in a smaller Rossby number under the cyclogeostrophic balance (Eq. 4), leading to an insignificant difference between the cyclogeostrophic and geostrophic flow velocities.(L210-212)

● The 'Discussion' section appears to be a continuation of results, focusing on case studies of ACE and CE in each region. Should a

separate section within the results be made called 'Case Studies'? I would not put new results in a discussion section.

**Response 2**: Thank you for your comments. The Results and Discussion sections have been integrated, and We have expanded the Discussion section to provide amore in-depth of analysis our research. (L555-562)

3.6 Case Studies

Statistical analysis of eddies in five North Pacific regions exhibiting significant differences between cyclogeostrophic and geostrophic EKE differences shows that streamline curvature substantially influences both fundamental eddy characteristics (such as lifespan, radius, formation, and dissipation) and key eddy parameters, including intensity, strain rate, and effective EKE. This influence is particularly strong in high-energy and low-latitude areas. In order to further explore the impact of streamline curvature on the evolution process of eddies, a comprehensive analysis was conducted on selected individual cases of both cyclonic and anticyclonic eddies from the SCS, KE, and HI regions.(L555-562)

● In the abstract there is mention of a few process changes but no actual discussion of this in the paper. Please consider what is written in the abstract should be discussed in the paper. For example, 'possess stronger potential energy'. I think this line refers to the 'Discussion' section and Figure 18 (?), but there is no discussion of this in the paper. I would disagree that the eddy possesses stronger potential energy. Is the eddy barotropic, is it baroclinic? The sea level anomaly does not change, and no subsurface examination is carried out, so it is hard to say this.

**Response 3**: Thank you for your comments. We have re-written the abstract. (L11-25)

The curvature of streamlines plays a significant dynamical and structural role in meandering currents. At scales comparable to the deformation radius, the motion of eddies is governed by a balance between the pressure gradient force, the Coriolis force, and the centrifugal force. For mesoscale eddies, the nonlinear term induced by the local curvature of streamlines is non-negligible. This study compares the statistical and dynamical parameters of mesoscale eddies under geostrophic and cyclogeostrophic

balances by examining five energetic North Pacific regions: the Aleutian Islands, Kuroshio Extension, South China Sea, California Coastal Current, and Hawaiian Islands. The comparison shows that cyclogeostrophic EKE is lower than geostrophic EKE for cyclonic eddies, whereas it is higher for anticyclonic eddies, particularly in energetic, low-latitude regions. The total number of eddies detected under the cyclogeostrophic balance is 35.65% lower than under the geostrophic balance. However, the frequency distributions of eddy radii in both frameworks show a right-skewed normal distribution. Detection under the cyclogeostrophic balance tends to eddies with larger radii and shorter lifespans. The velocity difference between the two balances for eddies increases with decreasing latitude. Similarly, case studies show that anticyclonic eddies exhibit more intense evolution under streamline curvature effects and are further prone to dissipation in low-latitude seas.

● There are a number of places throughout the paper that could benefit by further explanation. Some examples, 'eddy generation enhanced due to upwelling' (line 297), and 'AE influenced by curvature dissipate more readily' (line 607). Are there any process based studies that show these to be the case?

**Response 4**: Thank you for your comments.

(Line 297) To enhance the clarity of the findings, we have rewritten this sentence in the manuscript. (L294-296)

(Line 607) Regarding the five case studies, we have evaluated their necessity and streamline the discussion to focus on the most critical findings. Based on the statistical findings and the significance of streamline curvature, we have selected three regions — the Kuroshio Extension (KE), the South China Sea (SCS), and the Hawaiian Islands (HI)—for detailed case studies. (L563-633)

Furthermore, the description of some results has been rewritten to enhance accuracy.

The CC is one of the most productive eastern boundary upwelling systems and exhibits frequent mesoscale activity. However, due to the predominance of cyclonic circulation in this region, the number of cyclogeostrophic eddies is relatively lower than that of geostrophic eddies. (L294-296)

● It is interesting that the Kuroshio Extension has the largest eddy radii (Fig 5), could there be a reason for this?

**Response 5**: Thank you for your comments. Using satellite altimetry data, Ji et al., (2018) revealed two main eddy generation mechanisms in the KE region . Apart from barotropic instability, baroclinic processes leading to the meandering path of the KE, which generates eddies with large spatial scales and long lifetimes, dominate the second mechanism.

[Figure]

● The inclusion of the case studies is to try and draw some physical understanding from the results? Maybe this needs to be clear.

**Response 6**: Thank you for your comments. We have re-writern the inclusion.

● At the end of the Introduction section, you could outline your paper e.g. 'In section X we will discuss the results, and in section Y …'. This will solve my above point.

**Response 7**: Thank you for your comments. We have add this part in the

Introduction. (L96-101)

The structure of this paper is as follows: Section 2 presents the methods and data employed in the study.In Section 3, we examine the statistical characteristics of kinematic parameters, the differences in dynamical parameters of mesoscale eddies under cyclogeostrophic and geostrophic balance, as well as the contrasting dynamical processes of individual eddies between the two balances, are summarized in Section 4.

● Stegner and Dritschel (2000) [A Numerical Investigation of the Stability of Isolated Shallow Water Vortices] looked at the stability of ACE and CE in cyclogeostrophic balance; maybe this paper will help to add further support and locate more citations.

**Response 8:** Thank you for your suggestions, we have reviewed the references you provided and integrated them into our research framework. (L628-633)

While Stegner and Dritschel (2000) noted that centrifugal force in shallow-water models stabilizes anticyclones but destabilizes cyclones, Shakespeare (2016) demonstrated that anticyclones sharpen more rapidly than cyclones under identical background strain. In a complementary finding, Buckingham et al. (2021) diagnostically showed that streamline curvature and the Rossby number together limit anticyclonic vortex intensity. Consequently, for altimeter-derived surface eddies, the action of centrifugal force leads to more pronounced differences in anticyclonic structures.

● I somewhat understand the inclusion of strain rate and enstrophy, but feel it should probably be omitted. In 2D turbulence, dissipation is proportional to enstrophy, and so you could get some understanding from this. The strain rate similarly suggests dissipation but in 3D turbulence, which the 0.25 degree altimeter is never going to capture. Have there been studies that have looked at turbulence using satellite altimeter data?

They are probably heavily constrained by resolution. An alternative could be to include the wind stress curl and Ekman pumping, but you then need a wind stress data product… you could look at spatial changes in EKE?

**Response 9:** We sincerely appreciate the reviewer's comments, which are both insightful and highly constructive. **The strain rate** indeed plays a significant role in turbulent dissipation. Similar to vorticity, strain rate characterizes the kinematic state of an eddy (Yang et al., 2020). By combining surface drifters, satellite altimetry and satellite ocean-color data, Zhang et al., (2019) detect that when the strain rate of mesoscale surface geostrophic flow is strong, it favors a higher ageostrophic kinetic energy level and an increase in surface chlorophyll concentration. The strain-induced frontal processes are characterized by a surface chlor-ophyll increase and secondary ageostrophic upwelling along the light side of the oceanic density front.

The geostrophic strain rate $S_g$ is computed by the surface geostrophic velocity anomaly $(u_g, v_g)$ from the altimeter data:

$$S_g = \sqrt[2]{\left(\frac{\partial u_g}{\partial x} - \frac{\partial v_g}{\partial y}\right)^2 + \left(\frac{\partial v_g}{\partial x} + \frac{\partial u_g}{\partial y}\right)^2}$$

This formula has been included in the manuscript, along with a corresponding explanation. (L140-144)

Mesoscale deformation flow, often referred to as the geostrophic strain field, has been found to effectively enhance mesoscale frontal structures, regulate the development of ageostrophic disturbances in submesoscale

fronts, and play a significant role in controlling nutrient upwelling and enhancing marine primary productivity. Regions with the strongest strain rates correspond to areas of high eddy kinetic energy. Strain rate characterizes the deformation of the flow field, manifesting as stretching in one direction and compression in the perpendicular direction, continuously stretching and contracting in horizontal space, thereby disrupting the geostrophic balance of the flow. Consequently, areas with strong strain rates often induce frontal processes (McWilliams, 2016; Zhang et al., 2019).

[Figure]

**Schematic of surface flow strain rate.** Red vector arrows indicate the flow direction, black solid lines with arrows denote streamfunction contours, blue dashed lines represent the boundaries of fluid parcels before deformation, and blue solid lines show the boundaries after deformation.

**References :**

Yang, X., Xu, G., Liu, Y., Sun, W., Xia, C., and Dong, C.: Multi-source data analysis of mesoscale eddies and their effects on surface chlorophyll in the Bay of Bengal, Remote Sens., 12, 3485, https://doi.org/10.3390/rs12213485, 2020.

Zhang, Z., Qiu, B., Klein, P., and Travis, S.: The influence of geostrophic strain on oceanic ageostrophic motion and surface chlorophyll, Nat. Commun., 10, 2838, https://doi.org/10.1038/s41467-019-10883-w, 2019.

McWilliams, J. C. (2006). Fundamentals of geophysical fluid dynamics. Cambridge University Press.

● The results are often quite long winded, could they be broken up and focus given to the most significant finding?

**Response 10:** We appreciate the reviewer's valuable feedback. To improve clarity, we have revised the results section by dividing it into clearer subsections, streamlining the text, and emphasizing the key outcomes more prominently.

● Could other eddy census studies be discussed further e.g. Chelton et al (2011), Chen et al (2019) 10.1029/2019JC014983

**Response 11:** Thank you for your suggestion. We have reviewed the references you provided and integrated them into our research framework. (L592, L638, L666)

● How do you think your results would differ if you chose an alternative eddy detection algorithm? Would eddy radii change? Just something to think about.

**Response 12:** Thank you for this relevant question. We agree that the choice of eddy detection algorithm can influence statistical results, including total numbers and radius distributions of eddies. As shown in You et al. (2021), the GOMEAD dataset (used eddy detection algorithm presented in this study) contains 8% fewer eddies than META (dataset uses an SLA based automatic eddy detection algorithm). This discrepancy stems from their methodological differences: GOMEAD tends to underestimate eddy counts and lifespans near islands and filters out

eddies with small radii, while META may merge two eddies into one due to tracking jumps and misses eddies when sea level anomaly (SLA) signals are weak.

We focus on the differences between the cyclogestrophic corrected dataset and the original altimetry dataset, and have supplemented the discussion accordingly in the manuscript. (L266-267)

**Specific comments**

- Line 22-23: eddy kinetic energy and Eddy Kinetic Energy - decide what to use.

**Response 13:** Thank you for your comment. We have noted it and confirm that "eddy kinetic energy" is used in our manuscript. We have carefully checked and revised the terminology for consistency.

- Line 24: 'Eddies predominantly …' this isn't new.

**Response 14:** Thank you for your comments. We have re-written the abstract. (L11-25)

The curvature of streamlines plays a significant dynamical and structural role in meandering currents. At scales comparable to the deformation radius, the motion of eddies is governed by a balance between the pressure gradient force, the Coriolis force, and the centrifugal force. For mesoscale eddies, the nonlinear term induced by the local curvature of streamlines is non-negligible. This study compares the statistical and dynamical parameters of mesoscale eddies under geostrophic and cyclogeostrophic balances by examining five energetic North Pacific regions: the Aleutian Islands, Kuroshio Extension, South China Sea, California Coastal Current, and Hawaiian Islands. The comparison shows that cyclogeostrophic EKE is lower than geostrophic EKE for cyclonic eddies, whereas it is higher for anticyclonic eddies, particularly in energetic, low-latitude regions. The total number of eddies detected under the cyclogeostrophic balance is 35.65% lower than under the geostrophic balance. However, the frequency distributions of eddy radii in both frameworks show a right-skewed normal distribution. Detection under the cyclogeostrophic balance tends to eddies with larger radii and shorter lifespans. The velocity difference between the two balances for eddies increases with decreasing latitude.

Similarly, case studies show that anticyclonic eddies exhibit more intense evolution under streamline curvature effects and are further prone to dissipation in low-latitude seas.

- Line 27: 'Vorticity is more stable' - the eddy is more stable by virtue of a smaller Rossby number?

**Response 15:** Thank you for your comments. In order to present the research outcomes more clearly, we have re-written the abstract. (L11-25)

- Line 31-32: Where do you look at eddy-eddy and eddy-current interactions?

**Response 16:** Thank you for your comments. In order to present the research outcomes more clearly, we have re-written the abstract. (L11-25)

- Line 50: space between 'kilometers(Chelton'.

**Response 17:** Thanks for your careful checks. We are sorry for our carelessness. Based on your comments, We have added spaces. (L41)

- Line 76: 'where leeward… shows significant periods…' do you cycles/variability?

**Response 18:** Thank you for your question. We have re-writern this part. (L65-69)

The Hawaiian Islands region is recognized as an area of frequent eddy occurrence (Calil et al., 2008). Using available datasets of sea surface height, sea surface temperature, and surface wind stress, Yoshida et al. (2011) further investigated the interannual-to-decadal variability of eddies within the Hawaiian Lee Countercurrent (HLCC) band.

- Line 108: source for satellite data?

**Response 19:** Thank you for your review. We have reorganized this part. (L104-114) The data download URL has been included in the manuscript. (L699-701)

- Line 112: put 'data are accessible …' in data availability, and cite in text.

**Response 20:** Thank you for your suggestions. We have revised the relevant sections according to your suggestions.(L104-114)

The data download URL has been included in the manuscript. (L699-701)

- Line 121: what is the velocity fluctuation? Spatial, temporal - 5,10 days?

**Response 21:** Thank you for your question. The relevant explanations have been added to the manuscript. (L132-137)

The eddy kinetic energy (EKE) is computed at each point in space and time as::

$$EKE = \frac{1}{2} \times \left[ (u')^2 + (v')^2 \right] \tag{6}$$

where $u'$ and $v'$ are the surface velocity anomalies of zonal and meridional currents. The geostrophic eddy kinetic energy is obtained by the current components computed from the sea surface height anomalies (SSHA) of AVISO/DUCAS. For each component, the calculations are as follows:

$$u' = -\frac{g}{f} \left( \frac{\partial h'}{\partial y} \right) \tag{7}$$

$$v' = \frac{g}{f} \left( \frac{\partial h'}{\partial x} \right) \tag{8}$$

where $h'$ is the SSHA. The cyclogeostrophic eddy kinetic energy is obtained by the cyclogeostrophic surface currents.

- Line 123: 'percentage difference…' between geos and cylco?

**Response 22:** Thank you for your question. We have reorganized this part. (L138-146)

The normalized difference between climatic mean geostrophic EKE and cyclogeostrophic EKE is given by:

$$a_{EKE} = \frac{\overline{EKE_i} - \overline{EKE_g}}{\overline{EKE_g}} = \frac{\sum_{i=1}^{n} \frac{1}{2}\left(\left(u_i'\right)^2 + \left(v_i'\right)^2\right) - \sum_{i=1}^{n} \frac{1}{2}\left(\left(u_g'\right)^2 + \left(v_g'\right)^2\right)}{\sum_{i=1}^{n} \frac{1}{2}\left(\left(u_g'\right)^2 + \left(v_g'\right)^2\right)} \times 100\% \tag{9}$$

where $EKE_i$ denotes the EKE of the cyclogeostrophic surface currents, $EKE_g$ represents the EKE of the geostrophic surface currents. $u_i'$, $v_i'$ are the zonal and meridional velocity anomalies of the cyclogeostrophic surface currents, respectively, while $u_g'$ and $v_g'$ denote the corresponding velocity anomalies of the geostrophic surface currents.

● Eq 5 is the dynamical Rossby number.

**Response 23:** Thank you for your comment. We have modified Equation 5 to clarify its meaning. (L153-154)

The relative vorticity, which quantifies the rotational characteristics of an eddy, can be expressed as follows:

$$\zeta = \frac{\partial v'}{\partial x} - \frac{\partial u'}{\partial y} \tag{12}$$

● Line 166: 'andSCS'

**Response 24:** Thank for your careful checks. We are sorry for our carelessness. Based on your comments, we have restored the spaces in the text. (L196)

● Line 180: 'This study compares…' say this earlier?

**Response 25:** Thank you for this helpful suggestion. We have re-writtern this part. (L214)

● Line 183: 'Satellite altimetry overestimates…' using geostrophic assumption?

**Response 26:** Thank you for your sugesstion. We have revised this part.

Geostrophic EKE derived from altimetry overestimates actual EKE in cyclonic eddies and underestimates it in anticyclonic eddies.

● Line 190: 'Spatially, the difference…' could be reworded to clarify cyclo is more positive than geos over the domain?

**Response 27:** Thank you for your comment. We have taken your comments seriously and have made the required additional explanations in the revised version. (L219-220)

 indicating that the cyclogeostrophic current estimates exceed the geostrophic ones over most of the region

● Figure 1: Label each figure panel with the region (save reader from scanning caption and back to panels), remove legends from all but panel b.

**Response 28:** Thank you for the suggestion. We have re-plotted Figure1.

[Figure]

● Figure 2: Agulhas Insurgence?

**Response 29**:Thank you for your reminder. We sincerely thank the reviewer for careful reading. As suggested by the reviewer, we have corrected the 'Agulhas Insurgence ' into 'AI'. The abbreviation 'AI' is used here primarily because Figure 1 already employs 'Aleutian Islands (AI)'.

● Figure 2: Why are there differences across regions?

**Response 30**:Thank you for your comments. The variations in eddy

kinetic energy (EKE) evolution across regions in Figure 2 are rooted in the distinct dynamical regimes and generation mechanisms specific to each area. The rate of energy gain, peak intensity, and decay rate of an eddy are all primarily controlled by the physical process that formed it and the environmental conditions it experiences.

We have revised this paragraph to better emphasize the main points. (L250-267)

- Figure 3: Regions are quite small, can these be made bigger?

**Response 31:** Thank you for your suggestion. Figure 3 primarily compares the total number of eddy detections, while the regional map illustrates their spatial differences. To enhance the clarity of the figure, we have changed the colormap and adjusted the numerical range of the colorbar.

[Figure]

● Line 360: Any studies showing why these eddies are larger? Can you cite any.

**Response 32:** Thank you for your question. We have added relevant references in the manuscript. (L362)

**Such as:**

The study by Dong et al. (2007) demonstrates that in deep-water island wakes, the horizontal scale of eddies is controlled by the Burger number $Bu = (R_d / D)^2$, where $D$ is the island diameter and $R_d$ is the baroclinic deformation radius. For large islands such as Hawaii, $D \gg R_d(i.e., Bu \ll 1)$ In this regime: The eddy size is set by the larger baroclinic deformation radius $R_d$, not the island scale. Baroclinic instability becomes the dominant generation mechanism, preferentially producing vortices of scale $\sim R_d$. Thus, the low-Bu environment created by Hawaii's large

topography inherently favors the formation of mesoscale eddies with relatively larger radii.

Reference:

Dong, C., McWilliams, J. C., and Shchepetkin, A. F.: Island Wakes in Deep Water, J. Phys. Oceanogr., 37, 862–891, https://doi.org/10.1175/JPO3047.1, 2007.

Jia, Y., Calil, P. H. R., Chassignet, E. J., Metzger, E. J., Potemra, J. T., Richards, K. J., and Wallcraft, A. J.: Generation of mesoscale eddies in the lee of the Hawaiian Islands, J. Geophys. Res., 116, C11009, https://doi.org/10.1029/2011JC007305, 2011.

Yoshida, S., Qiu, B., and Hacker, P.: Wind-generated eddy characteristics in the lee of the island of Hawaii, J. Geophys. Res., 115, C03019, https://doi.org/10.1029/2009JC005417, 2010.

- Figure 9 and 10: Less eddies generated and less dissipated in cyclo. All regions are fairly similar, can the results be compressed into one figure, side by side with generation and dissipation? Feel free to change or not, just a suggestion. Could any other figures be condensed? Show total and then most significant finding?

**Response 33**:Thank you for your sugession. Figures 9 and 10 have been combined into a single figure, and the associated text has been revised accordingly. (L415-431)

Furthermore, Figures 15 and 16 have been combined into a single figure.

- Line 468: Why are there less eddies generated and dissipated under cyclo balance?

**Response 34**:Thank you for your review. Regarding the observation you mentioned that "the number of eddies generated and dissipated under cyclogeostrophic (CGEO) balance is lower than under geostrophic (GEO) balance,"this can be explained as follows: Under the geostrophic

assumption, the velocity field of eddies is derived solely from the balance between the pressure gradient and the Coriolis force, neglecting centrifugal force. In contrast, under cyclogeostrophic balance, centrifugal force is incorporated into the equilibrium equations. This modifies the velocity distribution, boundary shape, and intensity of eddies, causing some eddies to no longer satisfy vortex detection criteria. As a result, fewer eddies are detected under CGEO than under GEO. Additionally, CGEO correction may enhance the dynamic stability of eddies, prolong their lifespans, and make them more resistant to dissipation.

Here, we present several examples to illustrate the process of **Cyclogeostrophic correction:**

[Figure]

Correction of the eddy produced by shedding from the Kuroshio Large Meander.

[Figure]

The cyclogeostrophic correction process for shear-generated eddies.

[Figure]

The cyclogeostrophic correction process for eddies generated due to the combined effects of topography and the Kuroshio.

It is observed from the cases above that differential cyclogeostrophic correction at points of varying streamline curvature results in changes to

an eddy's shape and boundaries.

**A comparison of eddy boundaries before (GEO) and after (CGEO) correction:**

[Figure]

Reference:
Cao, Y., Dong, C., Stegner, A., Bethel, B. J., Li, C., Dong, J., ... & Yang, J. (2023).

Global sea surface cyclogeostrophic currents derived from satellite altimetry data. Journal of Geophysical Research: Oceans, 128(1), e2022JC019357.

Cao, Y., Dong, C., Qiu, Z., Bethel, B. J., Shi, H., Lü, H., & Cheng, Y. (2022). Corrections of mesoscale eddies and kuroshio extension surface velocities derived from satellite altimeters. Remote Sensing, 15(1), 184.

● Line 486: Fig 11b and d should be Fig 12?

**Response 35:** Thank you for your reminder. We have corrected it. Following the manuscript revisions, the figure order has been adjusted accordingly.

● Figure 12: More of a uniform dissipation of geos eddies in KE region but not in cyclo, why?

**Response 36:** Thank you for your question. In the cyclogeostrophic balance, the inclusion of centrifugal force means that the velocity field is determined by the equilibrium among the pressure gradient force, the Coriolis force, and the centrifugal force. This significantly alters the dynamic characteristics of eddies: anticyclonic eddies are notably strengthened, making them more resistant to dissipation, extending their lifespans, and thereby reducing their dissipation rates (Shakespeare, 2016). Cyclonic eddies, on the other hand, are weakened and may be stretched or broken down more quickly.

Meanwhile, in the Kuroshio Extension (KE) region, the eddy distribution is inherently polarized: the northern part is dominated by anticyclonic eddies, while the southern part is predominantly cyclonic (Itoh and Yasuda, 2010). Therefore, compared to the geostrophic (GEO) balance, the dissipation under CGEO appears non-uniform.

**Reference:**

Shakespeare, C. J.: Curved density fronts: Cyclogeostrophic adjustment and frontogenesis, J. Phys. Oceanogr., 46, 3193–3207, https://doi.org/10.1175/JPO-D-16-0137.1, 2016.

Itoh, S., and Yasuda, I.: Characteristics of mesoscale eddies in the Kuroshio–Oyashio Extension region detected from the distribution of the sea surface height anomaly, J. Phys. Oceanogr., 40, 1018–1034, https://doi.org/10.1175/2009JPO4265.1, 2010.

● Figure 13: Label left column CE and right column ACE? Seems the biggest fluctuations are in SCS, any idea why?

**Response 37:** Thank you for your suggestion. We have updated the figure reference from Figure 13 to Figure 12 in the manuscript. We have added explanations and cited relevant literature. (L483-487)

To clarify, in **Figure 12**, the **rows** correspond to the eddy types: the **first row** (panels a, b) represents **CEs**, and the **second row** (panels c, d) represents **AEs**. The columns represent the velocity components: the **left column** (panels a, c) shows **zonal speed**, and the **right column** (panels b, d) shows **meridional speed**.

The most significant speed fluctuations in the South China Sea (SCS) as shown in **Figure 12** can be attributed to a combination of the following factors:

**A. Complex Topography and Current Interactions:** The SCS features complex underwater topography, including islands, seamounts, and continental slopes, which disturb flow patterns and enhance spatial variability in eddy velocities.

**B. Strong Background Shear and Eddy Activity:** Influenced by monsoons and Kuroshio intrusions, the SCS exhibits strong background

current shear. Frequent eddy generation and eddy–current interactions further amplify velocity fluctuations.

**C. Low-Latitude Coriolis Effect:** Located at low latitudes, the SCS has a small Coriolis parameter, making nonlinear effects (e.g., centrifugal forces) more significant. This leads to more pronounced velocity differences after cyclogeostrophic (CGEO) correction.

These factors collectively explain why the SCS shows the largest speed variability among all studied regions.

● Figure 15: Could be consistent with labelling? E.g. KE region is sometimes presented first, but here it is in panel d. Why are there such differences, even in geos and cyclo, this is where other literature could come in use.

**Response 38:** Thank you for your guidance. To ensure consistency, we have standardized the order of regions across all figures in the manuscript. In the manuscript, the figures that have been revised are: Fig. 2, Fig. 3, Fig. 4, Fig. 8, Fig. 9, Fig. 12, Fig. 13 and Fig. 14.

● Figure 16: You could almost say the vorticity and strain are inversely related?

**Response 39:** Thank you for your suggestion. We agree with your perspective. In fluid dynamics, vorticity (describing local rotation) and strain rate (describing local stretching/shearing) collectively form the velocity gradient tensor, providing a complete kinematic description of a

fluid parcel's motion. From an energy perspective, they jointly partition the kinetic energy of turbulence or disturbances. Under certain conditions, an increase in one quantity may occur at the expense of the other.

Figures 15 and 16 have been combined into a single figure, and the associated text has been revised accordingly.

● Line 580: Change name of discussion? Do you need to include all five case studies here?

**Response 40:** Thank you for these suggestions.

The original discussion has been integrated with the results and placed within Section 3.6 ("Case Studies").

Regarding the five case studies, we have evaluated their necessity and streamline the discussion to focus on the most critical findings. Based on the statistical findings and the significance of streamline curvature, we have selected three regions一the Kuroshio Extension (KE), the South China Sea (SCS), and the Hawaiian Islands (HI)一for detailed case studies.

● Line 732: Could additional sources and references be included here to tie up your findings? What are the implications, bigger picture, takeaway for the reader?

**Response 41:** Thank you for your guidance, we have added relevant references based on your suggestion and cited them appropriately.We agree that explicitly stating the broader implications strengthens the

impact of our study. As suggested, we have added this part. The specific additions are marked highlight in the revised manuscript. (L635-697)

- Line 780: Where do you discuss eddy-eddy interactions and how do you define this?

**Response 42:** Thank you for your review comments. We have re-writern this part. (L683-685)